# Phosphorylation-mediated conformational change regulates human SLFN11

Michael Kugler [1,2], Felix J. Metzner[1,2], Gregor Witte [1], Karl-Peter Hopfner [1] & Katja Lammens [1] ✉

Human Schlafen 11 (SLFN11) is sensitizing cells to DNA damaging agents by irreversibly blocking stalled replication forks, making it a potential predictive biomarker in chemotherapy. Furthermore, SLFN11 acts as a pattern recognition receptor for single-stranded DNA (ssDNA) and functions as an antiviral restriction factor, targeting translation in a codon-usage-dependent manner through its endoribonuclease activity. However, the regulation of the various SLFN11 functions and enzymatic activities remains enigmatic. Here, we present cryo-electron microscopy (cryo-EM) structures of SLFN11 bound to tRNA-Leu and tRNA-Met that give insights into tRNA binding and cleavage, as well as its regulation by phosphorylation at S219 and T230. SLFN11 phosphomimetic mutant S753D adopts a monomeric conformation, shows ATP binding, but loses its ability to bind ssDNA and shows reduced ribonuclease activity. Thus, the phosphorylation site S753 serves as a conformational switch, regulating SLFN11 dimerization, as well as ATP and ssDNA binding, while S219 and T230 regulate tRNA recognition and nuclease activity.

SLFN11 is a member of the interferon-inducible Schlafen (Slfn) family of proteins which is involved in various cellular processes[1]. SLFN11 plays diverse cellular roles, including acting as an antiviral restriction factor, particularly against viruses like HIV-1, by targeting the host's translation machinery in a codon-usage-dependent manner. In this context, SLFN11's endoribonuclease activity degrades specific transfer RNAs (tRNAs) thereby blocking viral protein synthesis[2–4]. Beyond its role in antiviral defense, SLFN11 has the potential to serve as a predictive biomarker for DNA damage-related treatment in various cancer types[5–7]. Several studies have demonstrated that high expression of SLFN11 is associated with increased sensitivity to DNA-damaging agents, including replication, poly (ADP-ribose) polymerase (PARP) and topoisomerase inhibitors as well as platinum-based compounds[8–15]. Conversely, many tumors with low or absent SLFN11 expression exhibit resistance to these types of chemotherapeutics[16–18]. The mechanism by which SLFN11 contributes to cancer treatment chemosensitivity involves the irreversible block of stalled replication forks, which eventually leads to cell death[13,19–22]. During this process, SLFN11 is recruited to replication protein A (RPA)-coated single-

stranded DNA (ssDNA) regions and co-localizes at replication foci[20,21,23]. However, Boon et al. recently indicated that SLFN11 also triggers ribosome stalling in response to DNA damage, leading to a global inhibition of translation and ultimately to the induction of apoptosis[24]. This function was abolished in the endoribonuclease deficient SLFN11 E209A mutant suggesting that tRNA cleavage activity is critical for induction of apoptosis[24].

In addition, recent research by Zhang and colleagues identified SLFN11 as a pattern recognition receptor (PRR) for immunostimulatory ssDNA, containing CGT/A motifs[25]. In this context, SLFN11 directly binds to CGT/A motif-containing ssDNA, leading to its translocation to the cytoplasm and subsequent activation of cytokine expression and cell death. The activation of the innate immune response in this pathway is dependent on SLFN11's tRNA cleavage activity.

As a subgroup III Slfn family member, human SLFN11 has a tripartite domain architecture consisting of an N-terminal nuclease domain, followed by a linker domain and a C-terminal helicase domain (Fig. 1a and Supplementary Fig. 1). Structural and functional studies revealed that SLFN11 assembles as a dimer[26]. The N-terminal domains

[1]Gene Center and Department of Biochemistry, Ludwig-Maximilians-Universität München, Feodor-Lynen Straße 25, 81377 Munich, Germany. [2]These authors contributed equally: Michael Kugler, Felix J. Metzner. ✉e-mail: klammens@genzentrum.lmu.de

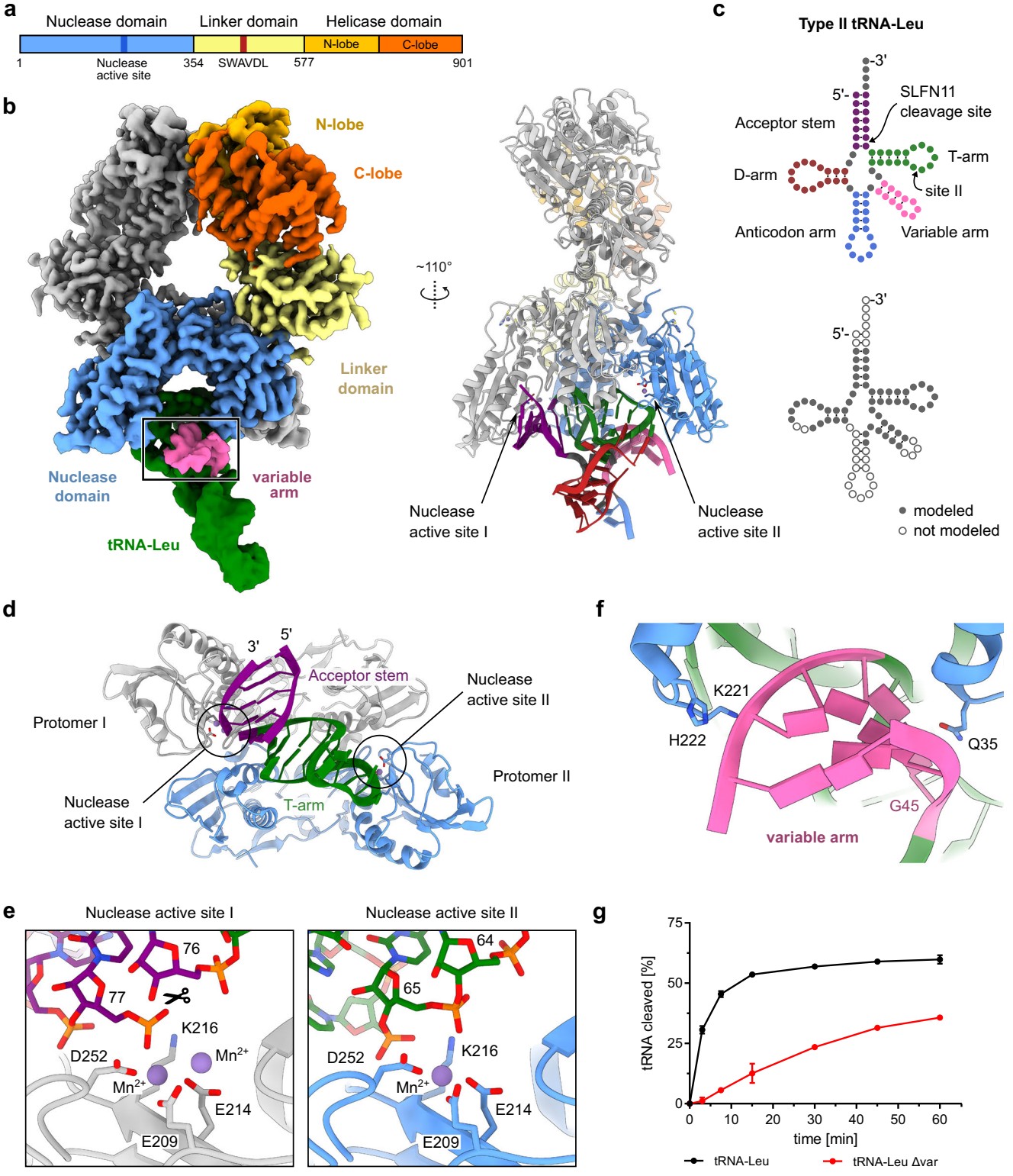

**Fig. 1 | Structural insights into tRNA-Leu recognition, binding, and cleavage by SLFN11. a** Domain architecture of SLFN11 with indicated key functional features. **b** Cryo-EM density map for SLFN11 bound to tRNA-Leu (left). tRNA is colored in green with the variable arm highlighted in pink. The position of the inset **f** is highlighted. The map was postprocessed in cryoSPARC by DeepEMhancer[44]. Cartoon representation of SLFN11 bound to tRNA-Leu with highlighted structural features (right). tRNA-Leu is color-coded based on the scheme in **c**. The nuclease active sites of both SLFN11 protomers are indicated. **c** Schematic representation of type II tRNA-Leu. Structural features of the tRNA are labeled and color-coded. The

SLFN11 cleavage site is indicated. **d** Bottom view of tRNA-Leu bound to the nuclease domains of the SLFN11 dimer. The nuclease active sites of both SLFN11 protomers are highlighted. **e** Close-up view on nuclease active site I (left) and II (right) with bound tRNA-Leu. The cleavage site of tRNA-Leu between position 76 and 77 is indicated in nuclease active site I. **f** Detailed view of the interaction of the nuclease domain of SLFN11 with the variable arm of tRNA-Leu. **g** Analysis of the endonucleolytic cleavage of tRNA-Leu and tRNA-Leu Δvar in a competitive tRNA-cleavage assay. Data are represented as mean values +/− SD from three independent experiments. Source data for **g** are provided as a Source Data file.

harbor its tRNA endoribonuclease active sites and the C-terminal helicase domains of the dimer are able to interact with ssDNA. These two domains are connected via a linker domain that includes the highly conserved SWAVDL motif. The endoribonuclease mechanism in the Slfn family of proteins has been studied by us[26,27] and others[28–32], yet insights into molecular recognition, binding, and cleavage of tRNA substrates are missing. Moreover, the functional mechanism of the helicase domain is enigmatic. The discrepancy between the inability of the SLFN11 dimer to bind and hydrolyze ATP[26], and the functional defect of the SLFN11 Walker B mutant in cells[21], remain an unresolved question.

Previously, it was shown that the enzymatic functions of SLFN11 are regulated by phosphorylation and dephosphorylation[29]. Mass spectrometry coupled with mutational analysis identified three functionally important SLFN11 phosphorylation sites. The respective alanine mutants maintained the ability to inhibit HIV p24 protein synthesis, while substitution with the phosphomimetic aspartic acid rendered the resulting proteins endoribonuclease inactive[29]. Based on these findings, it was proposed that the antiviral activity of SLFN11 is activated by dephosphorylation[29]. Further, it was found that treatment with DNA-damaging agents (DDAs) induces a reduction in SLFN11 phosphorylation, suggesting that dephosphorylation might also be connected to SLFN11's role in sensitizing cells to DDA treatment[29]. Recent results suggest that dephosphorylation of SLFN11 residue S753 is essential for chromatin localization, stalled replication fork blockage, and consequently, drug sensitivity to the DDA camptothecin[33]. There is evidence that the protein phosphatase 1 catalytic subunit gamma (PPP1CC) plays a role in dephosphorylating SLFN11, leading to its subsequent activation[29]. Additionally, the involvement of PP2A is suggested by the presence of a putative PP2A binding motif in SLFN11, which contributes to SLFN11-mediated drug sensitivity[31]. However, despite identifying the specific phosphatases of SLFN11, the corresponding kinases remain unknown.

All the functions of SLFN11 described above seem to depend on its endoribonuclease activity. To gain insights into SLFN11's tRNA substrate specificity, recognition, and cleavage, we solved cryo-EM structures of SLFN11 bound to tRNA-Leu and tRNA-Met and conducted biochemical analyses of its tRNA binding and endoribonuclease functions. These structures, along with biochemical data, also elucidate how the phosphorylation sites S219 and T230 in SLFN11 affect tRNA binding and cleavage and provide insights that may extend to other Slfn family members.

In order to understand the regulatory mechanism of SLFN11's diverse functions in molecular detail, we analyzed the SLFN11 phosphomimetic mutants with respect to their tRNA cleavage activity, nucleic acid binding and oligomeric state. The cryo-EM reconstruction of the phosphomimetic mutant SLFN11^S753D revealed a large conformational rearrangement of the helicase domain. Contrary to SLFN11^wt this mutant is monomeric and the helicase domain is rotated with respect to the N-terminal domain by approximately 140°, freeing the nucleotide binding site and allowing for ATP binding. In contrast, SLFN11^S753D lost its ability to bind ssDNA and, similar to SLFN11^S219D and SLFN11^T230D, showed reduced nuclease activity. Collectively, the data presented here explain the underlying molecular mechanism by which phosphorylation and dephosphorylation regulate SLFN11's diverse functions.

## Results
### Mechanism of tRNA binding and cleavage by SLFN11
To elucidate the mechanism of tRNA recognition by SLFN11, we aimed to determine the structures of SLFN11 bound to specific tRNAs. tRNAs can be classified into type I and type II based on the length of their variable regions, with type II tRNAs exhibiting longer variable arms. Previously, it was shown that SLFN11 binds both types of tRNAs in vitro, while preferentially cleaving type II tRNAs in cells[19]. Notably, SLFN11-

mediated attenuation of type II tRNA-Leu-TAA appears to be responsible for the observed translational inhibition of viral and host proteins[19]. Using a medium-resolution cryo-EM density, we previously showed that SLFN11 binds tRNA in a positively charged groove formed by the nuclease domains of the SLFN11 dimer[26]. To understand the tRNA binding mode in more detail and investigate the effect of the variable arm, we structurally analyzed SLFN11 in the presence of type II tRNA-Leu-TAA. The 3D reconstruction at a global resolution of 2.82 Å allowed us to model large parts of the tRNA molecule in a post-cleavage state (Fig. 1b, c and Supplementary Figs. 2 and 3a). The structure indicates that the SLFN11 nuclease domains interact with the acceptor stem and T-arm of the tRNA, primarily by coordinating the phosphate backbone (Fig. 1d). The tRNA is positioned close to both endonuclease active sites of the dimer, approximately 10 and 22 nucleotides from the 3' end of the tRNA, respectively (Fig. 1b, d). With the structure of SLFN11 bound to type II tRNA-Leu in hand, we analyzed the cleavage reaction of tRNA-Leu by SLFN11 in a nuclease cleavage assay. We measured tRNA cleavage over time for four different SLFN11 concentrations (Supplementary Fig. 4a). Differences in the extent of tRNA cleavage for different SLFN11 concentrations are most visible within the first 15 minutes of the reactions. Over time, the cleavage reactions are slowing down, as is evident from flattening of the curves (Supplementary Fig. 4a). We observed that all reactions reached a plateau after 45 minutes, yet none of the reactions reached a substrate conversion close to 100 %. The highest conversion was observed for 50 nM SLFN11, which cleaved 65 % of 50 nM tRNA-Leu after 60 minutes. This suggests that the endonuclease reaction may be hindered by product binding under the assay conditions, which is also supported by the fact that the cleaved tRNA remains bound to the protein in the cryo-EM structure. However, the cleavage assay performed at varying concentrations of SLFN11 suggests that more than one tRNA molecule is cleaved per SLFN11 dimer (Supplementary Fig. 4a). Additionally, a fraction of the tRNA might be misfolded, making it inaccessible for cleavage by SLFN11.

Based on the obtained cryo-EM density of the tRNA and biochemical data, tRNA-Leu is cleaved only at nuclease active site I between nucleotides 76 and 77 (Fig. 1e and Supplementary Fig. 4b). The cryo-EM density suggests the generation of 5'-phosphoryl and 3'-hydroxyl terminal ends. The cleavage position aligns with previous results mapping the SLFN11 endonucleolytic cleavage site for type II tRNA-Ser 10 nucleotides from the 3' end[26]. Compared to the SLFN11^wt apoenzyme (PDB: 7ZEL)[26] we observe only minor conformational changes of amino acid side chains in close proximity to nuclease active site I. In nuclease active site I, additional density indicates the presence of a second metal ion, whereas nuclease active site II shows density for only one metal ion (Fig. 1e and Supplementary Fig. 4b). The ions are coordinated by residues E209 and E214 and were assigned as manganese ions, as the ribonuclease activity of SLFN11 is manganese-dependent as previously observed by us[26]. In a second cryo-EM dataset of SLFN11 with tRNA-Leu-TAA, we were able to 3D classify the particles to separate pre- and post-cleavage states of the tRNA, resulting in two distinct reconstructions: one with intact tRNA and one with tRNA cleaved 10 nucleotides from the 3' end (Supplementary Figs. 5 and 6). In the pre-cleavage state, nuclease active site I shows density for two manganese ions, whereas the density for the second manganese ion vanishes in the post-cleavage state (Supplementary Fig. 6a). Furthermore, the position of the cleaved part of the tRNA (bases 1-10 from the 3' end) shifted slightly away from active site I (Supplementary Fig. 6b). Since only the cleavage-proficient active site I coordinates a second manganese ion, the cleavage reaction presumably follows a two-metal-ion catalytic mechanism (Fig. 1e and Supplementary Figs. 4b and 6a).

The non-catalytic SLFN11 protomer additionally interacts with the variable arm (nucleotides 45 to 56) via nuclease domain residues Q35, K221, and H222 (Fig. 1f). While most SLFN11-tRNA interactions are

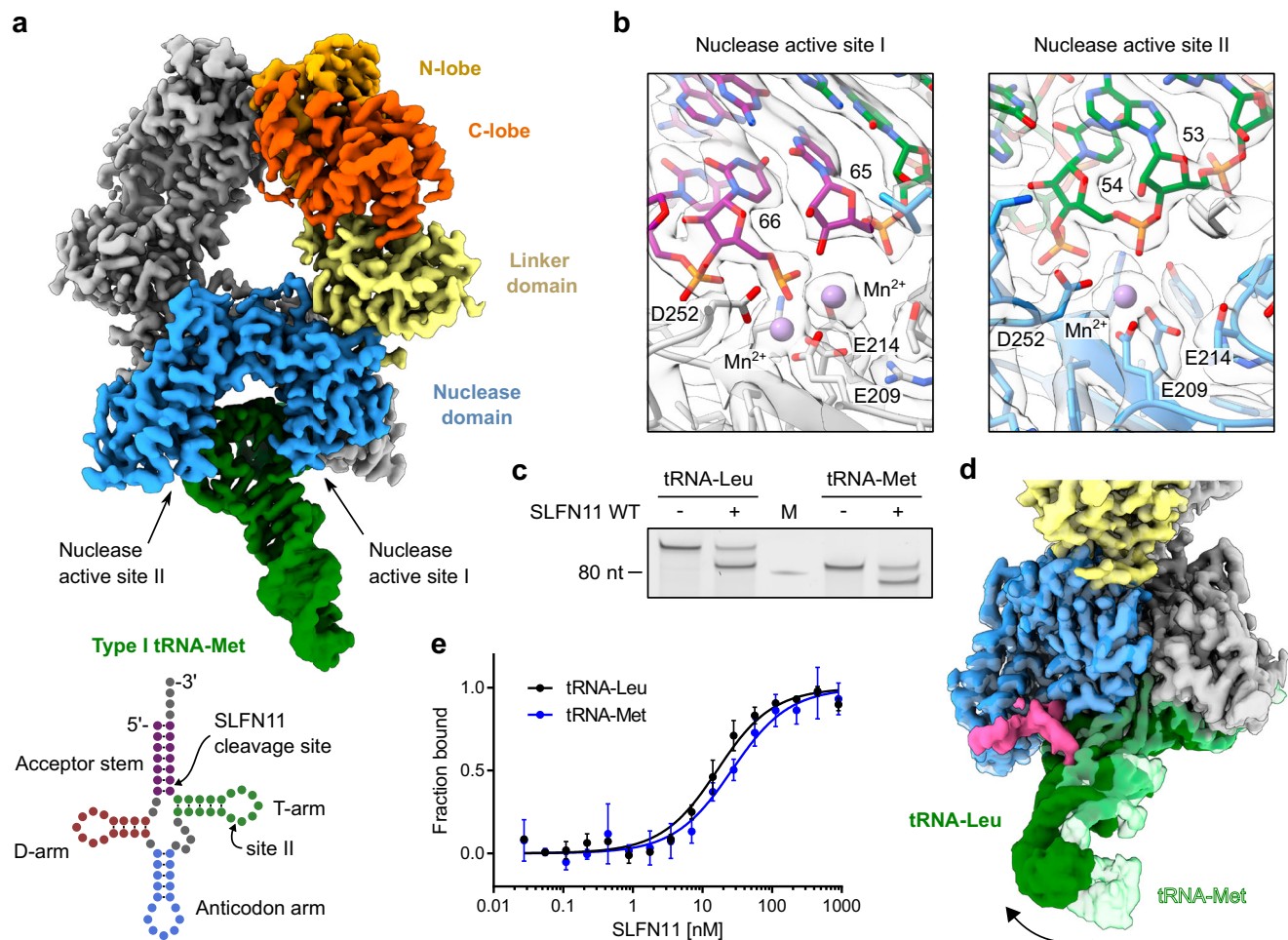

**Fig. 2 | Structural insights into tRNA-Met recognition, binding, and cleavage by SLFN11. a** Cryo-EM density map for SLFN11 bound to tRNA-Met (top). tRNA is colored in green and the nuclease active sites of both SLFN11 protomers are indicated. The map was postprocessed in cryoSPARC by DeepEMhancer[44]. Schematic representation of type II tRNA-Met (bottom). Structural features of the tRNA are labeled and color-coded. The SLFN11 cleavage site is indicated. **b** Close-up views of SLFN11 nuclease active sites I and II for the tRNA-Met bound structure. The semi-transparent cryo-EM map is shown together with the model. Nuclease active site residues (E209, E214, D252) and the manganese ions are labeled. **c** Endonuclease activity of SLFN11 on tRNA-Leu or tRNA-Met. The conditions of the nuclease assay correspond to the cryo-EM conditions. The experiment was performed in dupli-cates. One representative replicate is shown. **d** Difference in the position of tRNA-Leu and tRNA-Met relative to the nuclease domain of SLFN11. **e** MST measurements of SLFN11$^{wt}$ binding to fluorescently labeled type II tRNA-Leu and type I tRNA-Met to determine equilibrium dissociation constants ($K_d$). The indicated SLFN11 con-centrations assume a dimeric state. Data are represented as mean values +/− SD from three independent experiments. Source data for **c** and **e** are provided as a Source Data file.

electrostatic interactions between positively charged residues and the phosphate backbone of the tRNA, Q35 directly interacts with a guanine base (G45) of the variable arm, suggesting the possibility of a sequence readout (Fig. 1f and Supplementary Fig. 7a). Additionally, residue K253 of the nuclease domain interacts with guanine base (G17) of the D-arm that is highly conserved in type II tRNAs (Supplementary Fig. 7b, d). An AlphaFold[34] model of the tRNA-Leu-SLFN11 complex, extending beyond the density-based model, suggests that SLFN11 residues E225 and R229 could provide additional sequence readout of several bases of the tRNA acceptor stem (Supplementary Fig. 7c). Since the sequence of the variable arm and acceptor stem of type II tRNAs is little con-served, these contacts could play a role in specific tRNA recognition (Supplementary Fig. 7d). To further investigate the importance of the variable arm, we performed a competitive tRNA cleavage assay with tRNA-Leu (FAM-labeled) and tRNA-Leu lacking the variable arm (tRNA-Leu Δvar, Cy5-labeled) (Fig. 1g). We observed a reduction in tRNA cleavage activity upon deletion of the variable arm, highlighting the importance of this interaction (Fig. 1g).

To understand the differences between type I and type II tRNA binding modes, we determined the cryo-EM structure of SLFN11 bound to type I tRNA-Met-CAT (Fig. 2a, b and Supplementary Fig. 8). Under the conditions used for cryo-EM, tRNA-Met was cleaved 10 bases from the 3′ end by SLFN11, which is clearly visible in the cryo-EM density and confirmed by a nuclease gel of the respective sample (Fig. 2c). The cryo-EM reconstruction at a global resolution of 2.61 Å reveals similar binding modes and active site I and II geometries for both types of tRNA (Supplementary Fig. 3b). tRNA-Met is positioned in close proxi-mity to both endonuclease active sites of the dimer (Fig. 2a). The structure also contains two manganese ions in nuclease active site I, cleaving tRNA-Met between nucleotides 65 and 66, and a single man-ganese ion at nuclease active site II (Fig. 2b and Supplementary Fig. 4c). The conformation of tRNA-Leu and tRNA-Met is almost identical in the cleavage-deficient site II, though it slightly differs in nuclease active site I, including a slightly changed positioning of the second manga-nese ion (Supplementary Fig. 4c). As nuclease active site I is the cleavage-proficient site, the observed differences between tRNA-Leu and tRNA-Met might contribute to the cleavage specificity. A striking difference between the two structures is the interaction of the variable arm of tRNA-Leu with the nuclease domain of SLFN11. This interaction results in the tilting of tRNA-Leu towards SLFN11 compared to the

tRNA-Met bound structure (Fig. 2d). The interaction with the variable arm could allow SLFN11 to distinguish between type I and type II tRNAs, possibly affecting the affinity for different tRNAs. Hence, we performed Microscale Thermophoresis (MST) assays to calculate the binding constants of SLFN11 to both types of tRNA. However, under the conditions tested, and considering the assay's margin of error, SLFN11 binds to both tRNAs with comparable affinities. Specifically, tRNA-Leu had a $K_d$ value of $13.7 \pm 1.2$ nM, compared to $23.8 \pm 2.8$ nM for tRNA-Met considering dimeric SLFN11 (Fig. 2e). It was not possible to distinguish the individual binding constants of SLFN11 to cleaved or intact tRNA because, under the conditions of the MST measurement, the tRNA had already been partially cleaved (Supplementary Fig. 9).

## Regulation of SLFN11 functions by phosphorylation

Recently, three phosphorylation sites (S219, T230, and S753) were identified in SLFN11 that regulate its tRNase activity and its ability to sensitize cells to DDA treatment[29]. Two phosphorylation sites (S219, T230) are located in the nuclease domain, while the third site (S753) is located in the helicase domain (Fig. 3a, b). Residues S219 and T230 are in close proximity to the nuclease active site (Fig. 3a). Residue S753 is located in the region connecting the two ATPase lobes of the helicase domain, which is distant from the tRNA binding site (Fig. 3b). Based on the structure of the tRNA-SLFN11 complex, phosphorylation of residues S219 and T230 is likely to interfere with tRNA binding, due to repulsive forces of the negatively charged phosphate groups of the tRNA backbone and the phosphorylated residues S219 and/or T230 (Fig. 3a and Supplementary Fig. 10). Due to the tRNA binding mode by dimeric SLFN11, these phosphorylation sites might affect tRNA interactions in both protomers of the dimer, which could amplify their effectiveness (Fig. 3a and Supplementary Fig. 10). The phosphorylation site (S753) in the helicase domain is far from the tRNA binding site. On the other hand, it is located in close proximity to the ssDNA binding groove but does not directly interact with ssDNA (Fig. 3b)[26]. Thus, we decided to explore the effect of the phosphorylation sites individually on tRNA and DNA binding, as well as on the endonuclease activity of SLFN11. We mutated all three phosphorylation sites - S219, T230, and S753 - to phosphomimetic aspartates.

First, we compared the binding constants of SLFN11 phosphomimetic mutants to the wildtype protein using MST measurements with tRNA-Leu as the binding target (Fig. 3c). $K_d$ values could only be determined for SLFN11[wt], as all three mutants, including S753D in the helicase domain, exhibited strongly reduced binding to tRNA. Next, we evaluated the cleavage of tRNA-Leu by SLFN11 and the three phosphomimetic mutants in vitro. All three mutants showed a strong reduction in ribonuclease activity compared to SLFN11[wt] (Fig. 3d), which is consistent with their reduced tRNA-Leu binding capability. However, we did not observe a complete absence of nuclease activity as seen for previously characterized nuclease active site mutants[26]. SLFN11[S219D] showed the most dramatic reduction in tRNA cleavage with 8 % of the substrate cleaved after 60 minutes compared to 50 % cleaved by SLFN11[wt]. For SLFN11[T230D] and SLFN11[S753D] the substrate conversion at 60 minutes was reduced to around 25 %. Due to the close proximity of S219D and T230D to the tRNA and the nuclease active site, the observed reduction of nuclease activity is plausible (Fig. 3a and Supplementary Fig. 10). While residue S219 directly interacts with tRNA-Leu, residue T230 is positioned further from tRNA-Leu but in close proximity to the nuclease active site residues (Supplementary Fig. 10). This might explain why the nuclease cleavage is more strongly reduced for SLFN11[S219D] than for SLFN11[T230D]. As these residues are not directly participating in the cleavage reaction but are involved in tRNA recognition and binding, S219D and T230D show reduced nuclease activity rather than a completely abolished activity. However, there is no obvious structural explanation for the reduction in tRNA binding and endoribonuclease activity by the mutant SLFN11[S753D].

Further, DNA binding of the phosphomimetic mutants was analyzed by nano differential scanning fluorimetry (nanoDSF) and electrophoretic mobility shift assays (EMSA) (Fig. 3e, f and Supplementary Fig. 11). Both measurements showed that SLFN11[S219D] and SLFN11[T230D] bind to ssDNA but not to double-stranded DNA (dsDNA) (Fig. 3f and Supplementary Fig. 11). Thus, SLFN11[S219D] and SLFN11[T230D] exhibit the same DNA binding properties as SLFN11[wt], indicating that phosphorylation at these two residues does not affect SLFN11's ability to bind ssDNA. Strikingly, we observed that SLFN11[S753D] lost its ability to bind ssDNA (Fig. 3e, f). Nevertheless, like SLFN11[wt] and the other two phosphomimetic mutants, SLFN11[S753D] is also unable to bind dsDNA (Fig. 3e and Supplementary Fig. 11).

Finally, we compared nucleotide binding and hydrolysis capabilities of SLFN11[S219D], SLFN11[T230D] and SLFN11[S753D]. NanoDSF was used to probe binding to ATP and its analogues (Fig. 4a, b and Supplementary Fig. 12a). As previously reported for SLFN11[wt] [26], the addition of different nucleotides (ATP, ADP, and ATPγS) caused no changes in the inflection temperatures for SLFN11[S219D] and SLFN11[T230D], indicating no interaction with the tested nucleotides (Supplementary Fig. 12a). In contrast, the presence of ATP and ATPγS caused a large increase in the inflection temperatures of SLFN11[S753D], suggesting binding to these nucleotides (Fig. 4a, b). Despite its ability to bind ATP, we could not detect ATPase activity for SLFN11[S753D] alone or in presence of different DNA/RNA substrates or RPA (Supplementary Fig. 12b). This indicates that an additional factor like a specific substrate or an unknown interaction partner might be necessary to activate the ATP hydrolysis activity of SLFN11.

## Overall structure of SLFN11[S753D]

To understand the changes in the functional behavior of SLFN11[S753D] and to provide a structural basis for the regulatory role of the S753 phosphorylation, we employed cryo-EM to determine the structure of the phosphomimetic SLFN11 Ser 753 to Asp mutant bound to ATP (Fig. 4c and Supplementary Fig. 13). SLFN11[S753D] yielded exclusively monomeric 2D classes, which is in contrast to the predominantly dimeric state that was previously observed for SLFN11[wt] [26]. To counteract preferred particle orientation, cryo-EM data of SLFN11[S753D] were recorded at a tilt angle of 25°. This resulted in a 3D reconstruction of monomeric SLFN11[S753D] with a global resolution of 3.72 Å, allowing model building of residues 7-897 (Fig. 4c). Unlike in the monomeric SLFN11[E209A] reconstruction (EMD-14693)[26], the nuclease C-lobe of the SLFN11[S753D] reconstruction is resolved (Fig. 4c and Supplementary Fig. 14). The SLFN11[S753D] cryo-EM sample was prepared in the presence of ATP, which is in line with the observation of additional density in the nucleotide-binding pocket, suggesting a nucleotide bound state (Fig. 4c). Ab initio reconstructions of SLFN11[S753D] indicate different conformational states with respect to the helicase domain (Supplementary Fig. 13). In the presumably ATP-free reconstruction, only the helicase N-lobe is visible whereas both helicase lobes are ordered in the ATP-bound state (Fig. 4c and Supplementary Fig. 13).

ATP is coordinated by the Walker A and B motifs that are responsible for ATP binding and hydrolysis (Fig. 4a). The conserved glutamine residue of the Q-motif coordinates the adenine base. This is different from the autoinhibited conformation of the ATPase domain in SLFN11[wt] (PDB: 7ZEL)[26] where the Q-motif faces away from the nucleotide binding pocket (Fig. 4b).

To test whether the monomeric state of SLFN11[S753D] observed by cryo-EM is also present in solution, SLFN11[S753D] was analyzed by mass photometry (Fig. 4d, e and Supplementary Fig. 15). The analyses of all three phosphomimetic mutants showed that SLFN11[S219D] and SLFN11[T230D] exhibited a similar salt-sensitive monomer-dimer equilibrium as SLFN11[wt], indicating that these mutations do not affect dimerization (Supplementary Fig. 15). In contrast, SLFN11[S753D] was observed only in the monomeric state (104 kDa) in solution (Fig. 4d). This suggested that the phosphomimetic S753D mutant inhibits

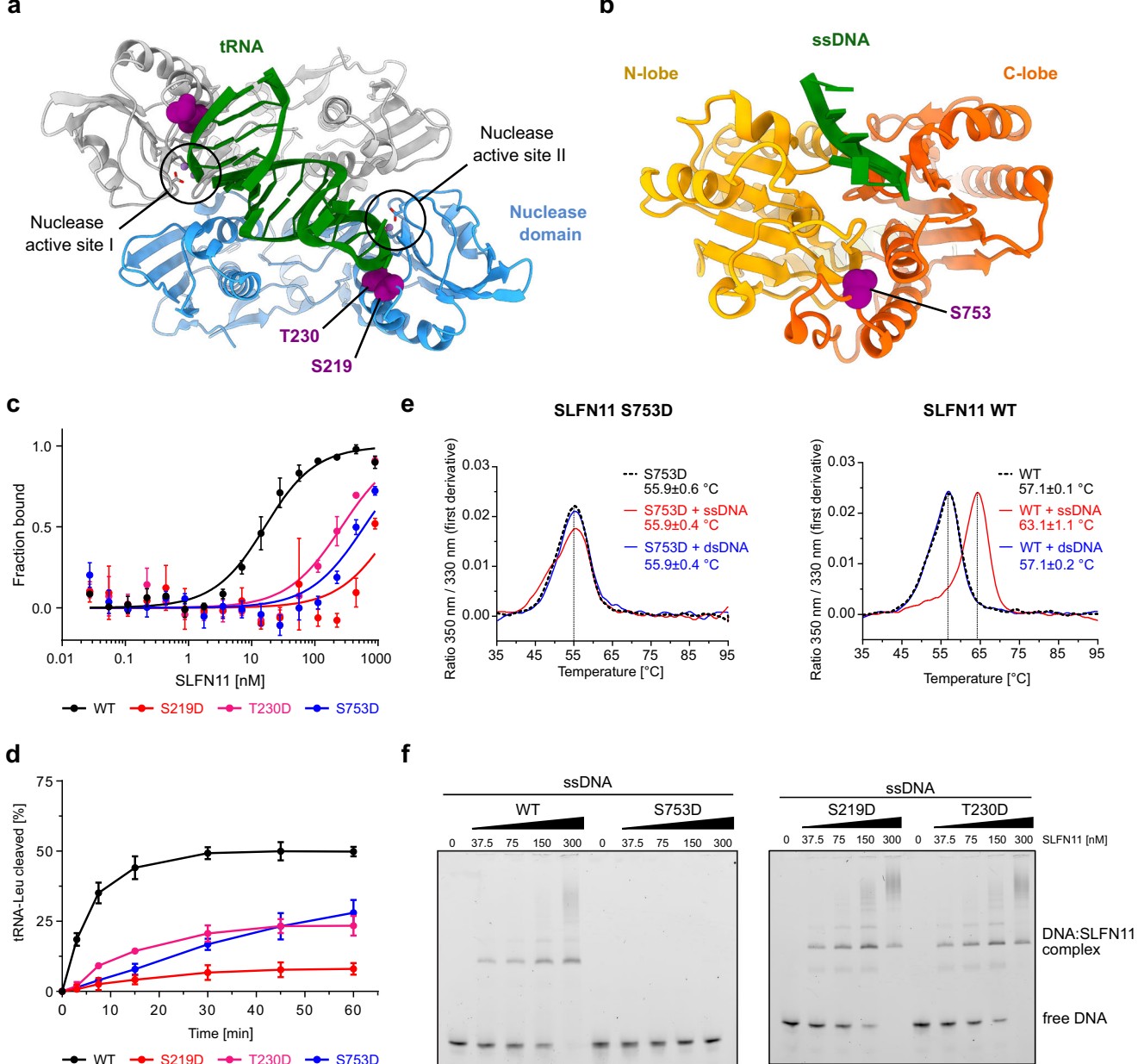

**Fig. 3 | Characterization of SLFN11 phosphomimetic mutants S219D, T230D, and S753D. a** Positions of phosphorylation sites S219 and T230 (purple) in the nuclease domains of SLFN11[wt] bound to tRNA-Leu. The nuclease active sites of both SLFN11[wt] protomers are indicated. **b** Position of phosphorylation site S753 (purple) in the ssDNA-bound helicase domain of SLFN11[wt] (PDB: 7ZES). **c** MST measurements of SLFN11[wt] and phosphomimetic mutants (S219D, T230D, S753D) binding to fluorescently labeled type II tRNA-Leu to determine equilibrium dissociation constants ($K_d$). The indicated SLFN11 concentrations assume a dimeric state. Data are represented as mean values +/− SD from three independent experiments. **d** Analysis of tRNA-Leu cleavage by SLFN11[wt], SLFN11[S219D], SLFN11[T230D], and SLFN11[S753D]. Data are represented as mean values +/− SD from three independent experiments. **e** NanoDSF measurements of SLFN11[S753D] (left) and SLFN11[wt] (right) in the presence of ssDNA and dsDNA. Data are represented as mean values +/− SD from three independent experiments. One representative replicate is shown. **f** ssDNA binding of SLFN11[wt] and SLFN11[S753D] (left), as well as SLFN11[S219D] and SLFN11[T230D] (right) monitored by electrophoretic mobility shift assays. The experiment was performed in duplicates. One representative replicate is shown. Source data for **c**–**f** are provided as a Source Data file.

SLFN11 dimerization, even though this residue is far away from the dimer interface (Figs. 3b and 4c).

### Structural comparison of SLFN11[wt] and SLFN11[S753D]

Structural comparison of SLFN11[wt] (PDB: 7ZEL)[26] and SLFN11[S753D] reveals large differences in the conformation of the helicase and linker domains (Fig. 5), especially within the interdomain (ID)-region (residues 560 to 591). In SLFN11[wt], this part adopts a long helical conformation, referred to as ID-helix, locking the helicase domain in an autoinhibited state (Fig. 5a)[26]. The same region adopts a vastly

different conformation in SLFN11[S753D], disrupting the long ID-helix, and forming a more open conformation (Fig. 5b). The conformational change of the ID-region between SLFN11[wt] and SLFN11[S753D] is accompanied by a rotation of the entire helicase domain by approximately 140°.

The loss of dimer interface I, as a result of the rotated helicase domains, likely destabilizes the dimeric state, explaining why SLFN11[S753D] adopts a solely monomeric state in solution (Supplementary Fig. 16). Both orientations of the helicase domains are stabilized by varying interactions with the linker domain (Fig. 5). In SLFN11[S753D] the

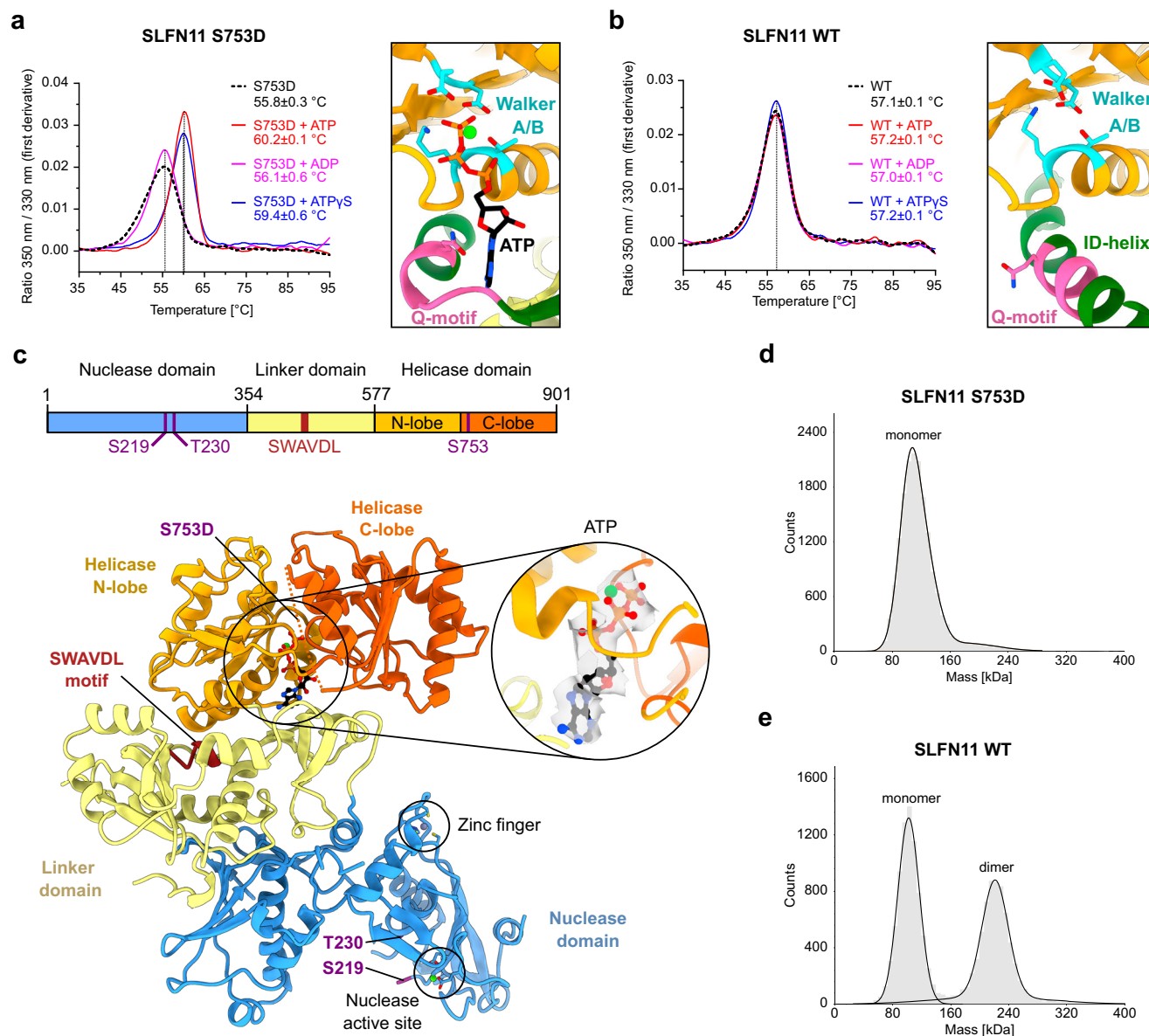

**Fig. 4 | Structure of phosphomimetic SLFN11^{S753D} in an ATP-bound state.**
**a** NanoDSF of SLFN11^{S753D} without or in the presence of different nucleotides. Data are represented as mean values +/− SD from three independent experiments. One representative replicate is shown. Detailed view of the nucleotide-binding region in the helicase domain of SLFN11^{S753D}. Walker motifs and Q-motif are highlighted in cyan and hot pink, respectively. ATP is colored in black and the magnesium ion is colored in green. **b** NanoDSF of SLFN11^{wt} without or in the presence of different nucleotides. Data are represented as mean values +/− SD from three independent experiments. One representative replicate is shown. Detailed view of the nucleotide-binding region in the helicase domain of SLFN11^{wt}. Walker motifs and Q-motif are highlighted in cyan and hot pink, respectively. **c** Domain architecture of

SLFN11 with indicated key functional features (top). The phosphorylation sites are highlighted in purple. Ribbon representation of the ATP-bound structure of SLFN11^{S753D} with highlighted structural features (bottom). The unresolved region from residue 747 to 771 is indicated by a dotted line. The approximate position of mutation S753D and the positions of S219 and T230 are indicated in purple. The inset shows the cryo-EM density map around ATP bound to the helicase domain of SLFN11^{S753D} (surface cutoff: 2 Å). The ATP-bound magnesium ion is colored in green. **d** Mass distribution of SLFN11^{S753D} in the presence of 80 mM NaCl observed by mass photometry. **e** Mass photometry analysis of SLFN11^{wt} in the presence of 80 mM NaCl showing a monomer-dimer mix. Source data for **a** and **b** are provided as a Source Data file.

ID-region, harboring the Q-motif, is involved in ATP binding. Residue F561 creates a hydrophobic contact with the N-lobe of the helicase domain and contributes to the formation of the nucleotide-binding pocket. Additionally, the N-terminal part of the linker domain forms a second interdomain region (ID-region II) that holds the C-lobe of the helicase domain in place (Fig. 5b). In contrast, the helicase domains in dimeric SLFN11^{wt} (PDB: 7ZEL)[26] are fixed in their position by dimer interface I and stabilized by the ID-helix, whereas residues F561 forms hydrophobic contacts with the SWAVDL motif (Fig. 5a). However, for this conformation the helicase dimer interface in not essential, as the

helicase domain in monomeric SLFN11^{E209A} (EMD-14693)[26] adopts the same conformation as the helicase domain in dimeric SLFN11^{wt} (PDB: 7ZEL)[26] (Supplementary Fig. 14). Thus, the conformation of the helicase domain of SLFN11^{S753D} is not the result of its monomeric state.

Along with the rotation of the helicase domain in SLFN11^{S753D}, conformational changes around the ssDNA binding site might sterically hinder DNA interactions (Supplementary Fig. 17). We did not observe density for the region between residues 747 to 771 in SLFN11^{S753D}, suggesting flexibility (Fig. 5b). This region connects the helicase N- and C-lobes and harbors residue S753 (Supplementary

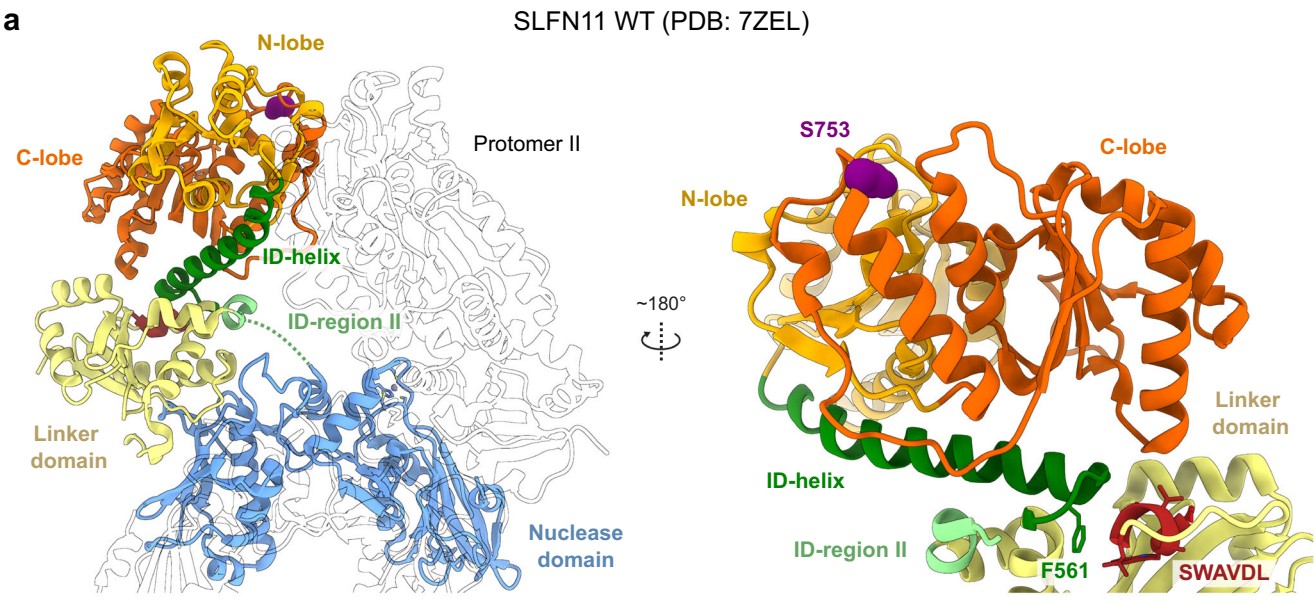

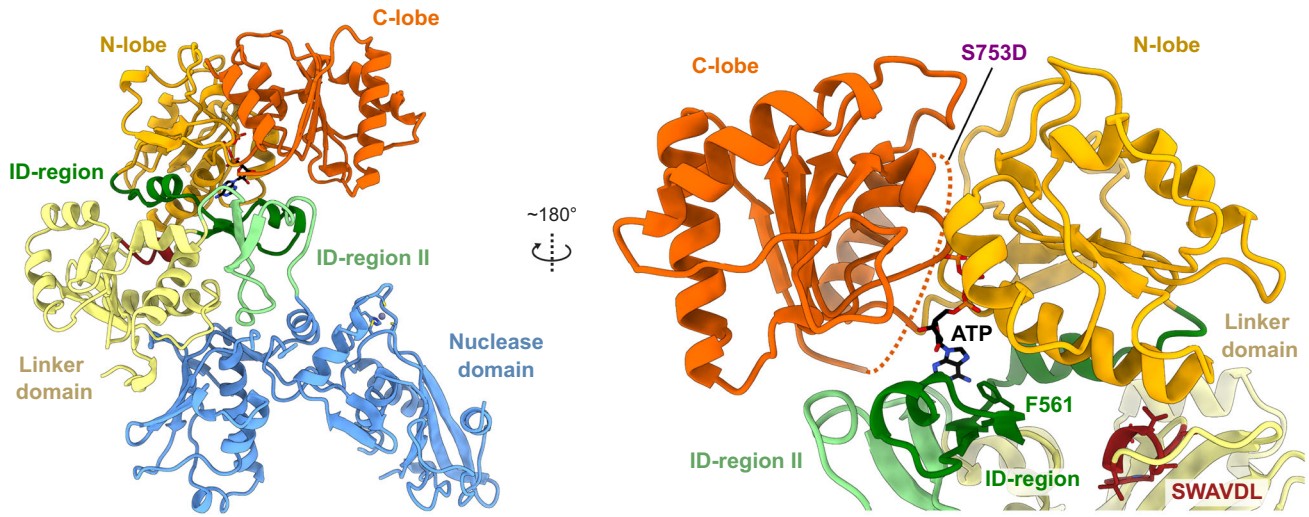

**Fig. 5 | Structural differences between SLFN11^wt and SLFN11^S753D. a** Ribbon representation of the SLFN11^wt dimer (PDB: 7ZEL) with highlighted structural features (left). The second protomer is depicted as transparent. The ID-helix is depicted in dark green and the ID-region II in light green. Detailed view of the SLFN11^wt helicase domain and the ID-regions (right). The position of residue S753 is indicated in purple. The hydrophobic interaction of residue F561 with the conserved SWAVDL motif (red) is shown. **b** Ribbon representation of SLFN11^S753D illustrating the newly formed interfaces of the rotated helicase domain with ID-regions I and II, colored in dark and light green, respectively (left). The bound ATP is shown in black. Detailed view of the SLFN11^S753D helicase domain and the ID-regions (right). The approximate position of mutation S753D (purple) is indicated.

Fig. 17a, b). Thus, the phosphomimetic S753D mutation destabilizes this region leading to conformational changes within the helicase domain. Two neighboring loop regions (residues 703 to 717 of the N-lobe and residues 871 to 880 of the C-lobe) undergo a concerted conformational change (Supplementary Fig. 17c, d). This results in a steric block of the ssDNA binding site by residues 871 to 880, when compared to ssDNA bound SLFN11 (PDB: 7ZES)[26] (Supplementary Fig. 17d). The conformational change might thus impede ssDNA binding by SLFN11^S753D.

Based on our findings, phosphorylation of S753, represented here by the phosphomimetic mutant SLFN11^S753D, triggers the rotation of the helicase domain by approximately 140°, as well as conformational changes within the linker and helicase domains, enabling nucleotide binding. Similarly, dephosphorylation of S753 locks SLFN11 in a nucleotide free state. Thus, besides affecting the oligomeric state of SLFN11, phosphorylation of S753 introduce another regulatory dimension by affecting nucleotide and ssDNA binding.

## Discussion

SLFN11 plays a crucial role in sensitizing cancer cells to DNA-damaging agents, acts as an antiviral restriction factor and as nuclear immune sensor for ssDNA. However, unregulated SLFN11 activity may prevent replication restart after ATR/CHK1-mediated transient replication blocks, preventing successful DNA replication. Also, uncontrolled cleavage of type II tRNAs by SLFN11 could negatively affect translation of endogenous transcripts. Therefore, SLFN11 requires tight regulation to avoid undesired SLFN11-mediated effects, like ribosome stalling and apoptosis in cells under stress-free or low-stress conditions.

Here, we provide a structural basis for tRNA binding and cleavage by SLFN11 and its regulations by phosphorylation. The structures of type I and type II tRNAs bound to SLFN11 help us to understand how SLFN11 is able to distinguish them on a molecular level. It especially showcases the importance of the variable arm in the recognition process (Fig. 1f and Supplementary Fig. 7). The possible sequence readout of the tRNA-Leu-TAA variable arm by SLFN11 suggests parallels to SLFN12, whose cleavage of type II tRNA-Leu-TAA was shown to be affected by the sequence of the variable arm[35]. The presence of the second manganese ion in the nuclease-proficient active site as well as the intact tRNA in the second active site proofs that dimeric SLFN11 incise the tRNA at a single cutting site, positioned 10 nucleotides from the 3′ end (Fig. 1e and Supplementary Fig. 4c). Our biochemical data show only minor differences in the binding affinity and cleavage reaction between type I and type II tRNAs in vitro (Fig. 2c, e). Therefore, this does not explain the specific ribosomal stalling at leucine-encoding UUA codons associated with the decreased abundance of tRNA-Leu-TAA observed in vivo[24]. However, the fact that the UUA codon and the corresponding tRNA-Leu-TAA are rather rare, might translate, together with the moderate cleavage preference for tRNA-Leu-TAA in vitro, to stronger effects in vivo. The tRNA-bound cryo-EM structures and nuclease assays suggest that the endonuclease reaction may be reduced due to product binding. This indicates that additional factors, such as tRNA modifications, play a role in determining specificity and enzymatic activity in cells. Additionally, the cleaved tRNA might be bound by the ribosome or other protein factors to facilitate its release from SLFN11. Thus, it remains to be investigated, whether ribosome stalling and cell death are a mere effect of reduced tRNA-Leu-TAA levels, or whether the tRNA-derived fragments themselves serve a specific function.

Phosphorylation of SLFN11 has been identified as an important regulatory posttranslational modification but the underlying molecular mechanism of SLFN11 regulation were not understood[29]. Comparison of the cryo-EM structure of SLFN11$^{S753D}$ with the structure of dimeric SLFN11$^{wt}$ shows that the phosphorylation status of S753 serves as a conformational switch of SLFN11 functions (Figs. 4, 5). The phosphorylated state, represented by the phosphomimetic SLFN11$^{S753D}$ mutant, is a monomer with a rotated helicase domain, that is stabilized by ID regions I and II. In this conformation, SLFN11$^{S753D}$ is unable to bind ssDNA, but is proficient in ATP binding, though not in hydrolysis and has diminished tRNA cleavage activity (Figs. 3–5). The observed reduction in the endonuclease activity of SLFN11$^{S753D}$ might be explained by its monomeric state, as we have shown that tRNA is bound to both protomers of the SLFN11 dimer and dimerization stimulates its endonuclease activity[26]. The conformational state of SLFN11$^{S753D}$ resembles the conformation of SLFN5$^{wt}$ (Supplementary Fig. 18). This finding demonstrates the similarity of SLFN11$^{S753D}$ with SLFN5, both of which are monomeric, stabilized by ATP binding and are incapable of ATP hydrolysis in vitro[26,27]. However, it has been shown that the Walker B motif of the ATPase, which is essential for ATP hydrolysis, is necessary for drug-induced cell killing and replication block by SLFN11[21]. This leads to the assumption that the ATP hydrolysis step is necessary for the function of SLFN11 in vivo. Nevertheless, the inability of SLFN11 to hydrolyze ATP independently, combined with the absence of double-stranded DNA-binding elements corresponding to SF1A helicase domains 1B and 2B, demonstrates that the protein alone is not a strand-opening helicase. Therefore, an additional factor, such as a binding partner, modification, or signal, is required to activate its ATPase and helicase activity.

In the unphosphorylated dimer state of SLFN11, the dimer interface, the ID-helix, as well as the F561-SWAVDL interaction, stabilize the helicase domain in an autoinhibited conformation (Fig. 5a). This SLFN11 conformation enables ssDNA binding and tRNA binding and cleavage, but renders SLFN11 incapable of binding to ATP. Since neither ATP nor ADP fit the nucleotide-binding pocket of the SLFN11 dimer, it must either be hydrolyzed or released to permit the conformational change from the ATP bound monomer to the dimer (Fig. 4b). The ATP exchange or hydrolysis cannot be explained by the current data, thus other factors, such as binding partners may play a role in vivo.

Dephosphorylation of SLFN11 may occur in a sequential manner, starting with the dephosphorylation of SLFN11$^{S753P}$ [29]. This initial step induces a conformational change that enables dimerization, activating SLFN11 from its dormant phosphorylated state. Fujiwara et al. supported the importance of the SLFN11 S753 phosphorylation site, as the constitutive phosphorylation mutant S753D abolished SLFN11-dependent drug sensitivity to camptothecin in human cells[33]. However, the tRNase activity of SLFN11 may still be inhibited at this point, due to phosphorylation of residues S219 and T230.

We present a model that combines our structural and biochemical findings with current literature, to illustrate the functional regulation of SLFN11 through phosphorylation (Fig. 6). Under normal, stress-free conditions, the nuclease and helicase domains of SLFN11 are phosphorylated at various sites (S219, T230, and S753), leading to the inhibition of dimerization, tRNA cleavage, and ssDNA binding. Consequently, SLFN11's ability to block stalled replication forks, bind immune stimulatory ssDNA, and exert its antiviral effects are suppressed. In response to replication stress induced by diverse DDAs, a critical stress threshold may be surpassed, inducing the dephosphorylation of SLFN11$^{S753p}$ by e.g. PPP1CC (Fig. 6a)[29]. By liberating the conformational inhibition, SLFN11 is enabled to dimerize, facilitating its interaction with stalled replication forks through its affinity for ssDNA. There, SLFN11 irreversibly blocks replication forks in an ATPase-dependent manner[21]. The stimulated endoribonuclease activity of dimerized SLFN11 toward certain type II tRNAs results in a reduced abundance of tRNA-Leu-TAA in both the nucleus and cytosol. This reduction causes ribosome stalling at UUA codons, triggering global translation inhibition through the integrated stress response (ISR) and inducing apoptosis via the ribotoxic stress response (RSR)[24].

Recently, it was reported that ssDNA comprising a CGT motif can activate SLFN11, triggering cytokine expression and subsequently leading to lytic cell death[25]. Hence, SLFN11 as well as the mouse homolog SLFN9 serve as immune sensors for ssDNA in a ssDNA sequence-dependent manner. Since the SLFN11$^{S753D}$ mutant is unable to interact with ssDNA, phosphorylation/dephosphorylation might control this recently identified innate immune sensor function (Fig. 6b). Dephosphorylation of SLFN11$^{S753p}$ induces a conformational change of the helicase domain, enabling SLFN11 to dimerize and interact with CGT motif containing ssDNA in the nucleus. It has been shown that recognition of ssDNA containing a CGT motif triggers SLFN11 translocation to the cytosol. Thus, dimerized SLFN11, bound to ssDNA, moves from the nucleus to the cytosol, where it cleaves type II tRNAs, ultimately leading to cell death[25]. This finding appears to resolve the conundrum of SLFN11 being a nuclear protein, while tRNA cleavage presumably occurs mainly in the cytoplasm. Additionally, this links the bipartite functions of ssDNA recognition in the helicase domain and tRNA cleavage in the N-terminal nuclease domain.

Residue S753 is not conserved among human subgroup III Slfn proteins and is exclusively found in SLFN11, while other human SLFN proteins have a proline at this position (Supplementary Fig. 1). This suggests that the regulation of SLFN11 through phosphorylation, which modulates its affinity for DNA and its oligomeric state, may be unique to SLFN11. This inhibitory mechanism could serve as a safeguard to prevent replication fork blockage and enable DNA repair under normal cellular conditions. In contrast, residue S219 is conserved in SLFN5, SLFN11, and SLFN13, with SLFN14 harboring a threonine at this position, indicating a potential common phosphorylation site for subgroup III Slfn members (Supplementary Fig. 1). Finally, T230 is unique to SLFN11, as it is not conserved among other subgroup III Slfn proteins (Supplementary Fig. 1). Phosphorylation of residues S219

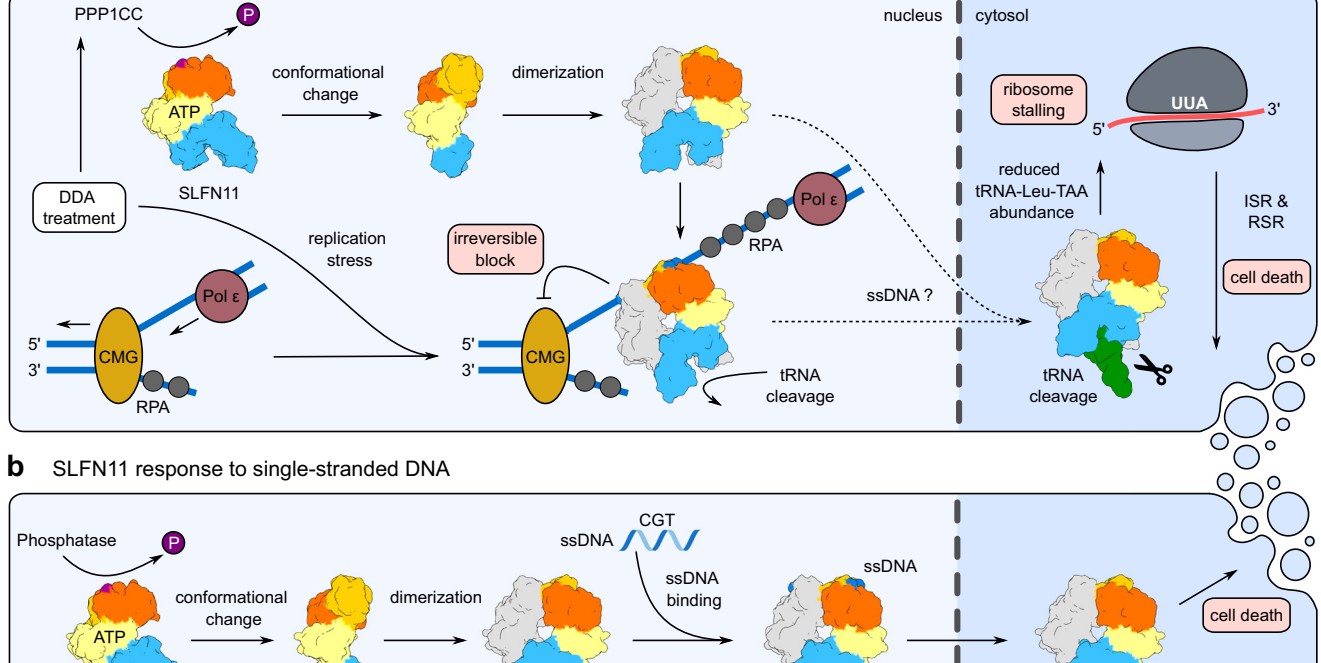

**Fig. 6 | Model for the activation of SLFN11 by DDAs and nuclear ssDNA.**
**a** SLFN11's response to DNA-damaging agents, leading to an irreversible block of stalled replication forks, ribosome stalling, and cell death. **b** Regulation of SLFN11's function as an immune sensor for nuclear ssDNA containing a CGT motif. The model was created based on our work[26] and the work of other groups[21,24,25,29].

and T230 can serve as an additional layer of regulation to control the tRNAase activity of SLFN11.

In summary, we provide a detailed molecular mechanism for tRNA recognition and cleavage by SLFN11 and its regulation by phosphorylation. With its diverse cellular functions, SLFN11 is a unique member of the Slfn family of proteins and understanding its regulatory mechanisms could open up possibilities for developing SLFN11-specific activators or inhibitors.

## Methods

### Protein expression and purification

SLFN11 expression and purification has been performed based on the previously published protocol[26]. Briefly, SLFN11 was cloned into the pFASTBac1 expression vector using Gibson assembly for expression in insect cells[36]. *Spodoptera frugiperda* Sf21 insect cells (Thermo Fisher) were used for virus generation. Expression was carried out in *Tricho-plusia ni* High Five cells (Invitrogen) at 27 °C and 95 rpm for 72 h. After 72 h, the cells were harvested by centrifugation, resuspended in lysis buffer (50 mM Tris pH 7.5, 400 mM NaCl, 2 mM MgCl$_2$) supplemented with protease inhibitors (0.18 g l$^{-1}$ PMSF, 0.32 g l$^{-1}$ benzamidine, 1.37 mg l$^{-1}$ pepstatin A, 0.26 mg l$^{-1}$ leupeptin, 0.2 mg l$^{-1}$ chymostatin) and disrupted by sonication. The lysate was cleared by centrifugation at 30,000 g at 4 °C for 45 min and the supernatant was incubated with pre-equilibrated ANTI-FLAG M2 Affinity Gel (Sigma-Aldrich) for 90 min. The resin was washed with wash buffer (25 mM Tris pH 7.5, 250 mM NaCl, 2 mM MgCl$_2$). After washing with buffer A (25 mM Tris pH 7.5, 120 mM NaCl, 2 mM MgCl$_2$, 1 mM DTT), the protein was eluted iteratively for five times in elution buffer (buffer A supplemented with 0.2 mg ml$^{-1}$ Flag-peptide) over 60 min. The eluate was loaded onto a HiTrap Heparin HP column (GE Healthcare) and the protein was eluted by a linear salt gradient (100 % buffer A to 100 % buffer B (25 mM Tris pH 7.5, 1 M NaCl, 2 mM MgCl$_2$, 1 mM DTT) over 12 CV). The peak fractions were combined and flash-frozen in liquid nitrogen.

SLFN11 mutants were prepared by quick change PCR and expressed and purified as wild-type protein. RPA used for ATPase assays has been expressed and purified as described previously[26].

### Cryo-EM sample preparation

Purified SLFN11$^{wt}$ was dialyzed overnight into cryo-EM buffer (20 mM HEPES pH 7.5, 80 mM NaCl, 2 mM MgCl$_2$, 1 mM DTT). The samples were diluted to a final concentration of 1.5 µM SLFN11$^{wt}$ and the NaCl concentration was adjusted to 100 mM. tRNA (either tRNA-Leu or tRNA-Met) was added to reach a final concentration of 3 µM. Subsequently, MnCl$_2$ was added to a final concentration of 2 mM and the samples were incubated at room temperature for 10 minutes. 4.5 µl of the samples were applied onto a glow-discharged QUANTIFOIL® R2/1 + 2 nm carbon Cu200 grid. The samples were vitrified in liquid ethane using an EM GP plunge freezer (Leica, 10 °C and 95% humidity).

Purified SLFN11$^{S753D}$ was diluted in cryo-EM buffer (25 mM Tris-HCl pH 7.5, 2 mM MgCl$_2$, 1 mM DTT) to a final concentration of 4 µM. ATP was added to a final concentration of 1 mM and the sample was incubated for 10 min on ice. 4.5 µl of the sample was applied onto a glow-discharged QUANTIFOIL® R2/1 Cu200 grid. The sample was vitrified in liquid ethane using an EM GP plunge freezer (Leica, 10 °C and 95% humidity).

## Cryo-EM data collection

Cryo-EM data of SLFN11[wt] bound to tRNA-Leu and SLFN11[S753D] were collected using a FEI Titan Krios G3 transmission electron microscope (300 kV) equipped with a GIF quantum energy filter (slit width 20 eV) and a Gatan K2 Summit direct electron detector (software used: EPU 3.5.1, TEM User interface Titan 3.15.1, Digital Micrograph 3.22.1461.0). Cryo-EM data of SLFN11[wt] bound to tRNA-Met and SLFN11[wt] bound to tRNA-Leu in pre-cleavage and post-cleavage state were collected using a FEI Titan Krios G3 transmission electron microscope (300 kV) equipped with a Selectris X imagining filter (slit width 10 eV) and a Falcon 4 direct electron detector (software used: EPU 3.5.1, TEM User interface Titan 3.15.1).

For the structure determination of SLFN11[wt] bound to tRNA-Leu 2546 movies were collected with a total electron dose of 41.2 e$^-$ Å$^{-2}$, fractionated into 40 movie frames over 8 s. The dataset was collected with defocus values ranging from −1.1 to −2.9 μm and a pixel size of 1.049 Å. For the structure determination of SLFN11[wt] bound to tRNA-Met 11,389 movies were collected with a total electron dose of 40 e$^-$ Å$^{-2}$, fractionated into 40 movie frames over 2.78 s. The dataset was collected with defocus values ranging from −0.5 to −2.6 μm and a pixel size of 0.727 Å. For the structure determination of SLFN11[wt] bound to tRNA-Leu in pre-cleavage and post-cleavage state, 11,778 movies were collected with a total electron dose of 40 e$^-$ Å$^{-2}$, fractionated into 40 movie frames over 2.78 s. The dataset was collected with defocus values ranging from −0.5 to −2.6 μm and a pixel size of 0.727 Å. For the structure determination of SLFN11[S753D] 7,480 movies were collected at a tilt angle of 25° with a total electron dose of 40.21 e$^-$ Å$^{-2}$, fractionated into 40 movie frames over 8 s. The datasets were collected with defocus values ranging from −1.1 to −2.9 μm and a pixel size of 1.045 Å.

## Cryo-EM image processing

Movie frames were motion corrected using MotionCor2 1.4.5[37]. All subsequent cryo-EM data processing steps were carried out using cryoSPARC 4.4.1[38]. The CTF parameters of the datasets were determined using patch CTF estimation (multi). The masks for the calculation of FSC were created by cryoSPARC automasking. The resolutions reported are calculated based on the gold-standard Fourier shell correlation criterion (FSC = 0.143) with 3D FSC plots using Remote 3DFSC Processing Server[39]. The exact processing schemes are depicted in Supplementary Figs. 2, 5, 8, and 13. The data collection and refinement statistics are summarized in Supplementary Table 1.

For the SLFN11[wt] bound to tRNA-Leu dataset, particles were initially picked on 2,546 micrographs using a blob picker (Supplementary Fig. 2). The particles were subjected to 2D classification, ab-initio reconstruction, and heterogeneous refinement. The particles in the class with clearly defined features were selected and used as input for Topaz train[40,41]. The resulting topaz model was used as a template for particle picking, where particles were extracted with a box size of 320 px and a pixel size of 1.049 Å. The particles were subjected to 2D classification and heterogeneous refinement yielding 1,776,105 particles. The particles in the class with clearly defined features were subjected to non-uniform refinement[42], 3D classification, and heterogeneous refinement. This process resulted into a clear separation of particles into tRNA-bound and tRNA-free class. The final resolution of SLFN11[wt] bound to tRNA-Leu reconstruction after non-uniform refinement[42] was 2.89 Å containing 762,328 particles. The sharpened map of this reconstruction was used further for model building and refinement. The final reconstruction after non-uniform refinement was subjected to 3D Flexible Refinement (3DFlex)[43]. Obtained Flex volume from 3D Flex Reconstruction was postprocessed by DeepEMhancer[44]. This map was only used for the preparation of the figures.

For the SLFN11[wt] bound to tRNA-Leu in pre- and post-cleavage state, particles were initially picked on 11,778 micrographs using a blob picker (Supplementary Fig. 5). Reasonable 2D classes were selected and used as input for Topaz train[40,41]. The resulting topaz model was

used as a template for particle picking and particles were subjected to ab-initio reconstruction and heterogeneous refinement. This yielded 8,360,668 particles extracted with a box size of 400 px and a pixel size of 0.727 Å. The yielded class of 2,072,829 particles with clearly defined features was used for further processing. These particles were further sorted by rounds of heterogeneous refinements, ab-initio and 3D classifications. Rounds of 3D classifications allowed the separation of two states of bound tRNA-Leu: a pre-cleavage state with uncleaved tRNA and a post-cleavage state with cleaved tRNA. The final resolution of SLFN11[wt] bound to tRNA-Leu in the pre-cleavage state after non-uniform refinement[42] was 3.00 Å containing 57,414 particles. The final resolution of SLFN11[wt] bound to tRNA-Leu in the post-cleavage state after non-uniform refinement[42] was 2.82 Å containing 114,044 particles. The sharpened maps of these reconstructions were used further for model building and refinement.

For the SLFN11[wt] bound to tRNA-Met dataset, particles were initially picked on 11,389 micrographs using blob picker (Supplementary Fig. 8). Reasonable 2D classes were selected and used as input for Topaz train[40,41]. The resulting topaz model was used as a template for particle picking and particles were subjected to ab-initio reconstruction and heterogeneous refinement. This yielded 8,912,516 particles extracted with a box size of 400 px and a pixel size of 0.727 Å. The yielded class of 4,112,708 particles with clearly defined features was used for further processing. These particles were further sorted by rounds of ab-initio, heterogeneous refinements, 2D and 3D classifications. The final resolution of SLFN11[wt] bound to tRNA-Met reconstruction after non-uniform refinement[42] was 2.64 Å containing 213,920 particles. The sharpened map of this reconstruction was used further for model building and refinement. The final reconstruction after non-uniform refinement was subjected to 3D Flexible Refinement (3DFlex)[43]. Obtained Flex volume from 3D Flex Reconstruction was postprocessed by DeepEMhancer[44]. This map was only used for the preparation of the figures.

For the SLFN11[S753D] dataset, particles were initially picked on 3,668 micrographs (25° tilt angle) using blob picker (Supplementary Fig. 13). Reasonable 2D classes were selected and used as input for Topaz train[40,41]. The resulting topaz model was used as a template for particle picking yielding 3,348,870 particles extracted with a box size of 256 px and a pixel size of 1.045 Å. The particles were subjected to 2D classification, ab-initio reconstruction, and heterogeneous refinement and the class with clearly defined features was selected (820,178 particles). An analogous processing strategy was used for two additional tilted datasets (2,672 micrographs and 1,140 micrographs, 25° tilt angle). The obtained particles from two additional tilted datasets (975,818 particles and 315,423 particles) were combined with particles from the first dataset (820,178 particles) and further sorted by heterogeneous refinement resulting in five classes with the two major classes. One of the class was ATP-free reconstruction with unresolved helicase domain C-lobe, whereas in the ATP-bound reconstruction both helicase lobes were ordered (Supplementary Fig. 13). The ATP-bound class, that showed the most defined features of SLFN11[S753D], was selected (672,274 particles) and used for further refinement steps. The final resolution of the SLFN11[S753D] reconstruction after non-uniform refinement[42] was 3.72 Å containing 239,084 particles. The sharpened map of this reconstruction was used further for model building and refinement.

## Model building and refinement

Our previously obtained structure of SLFN11 dimer apoenzyme (PDB: 7ZEL) was used as the starting model for both SLFN11[wt] tRNA-bound structures (tRNA-Leu and tRNA-Met). The coordinates of 7ZEL were rigid body docked into the cryo-EM density of SLFN11[wt] bound tRNA-Leu and tRNA-Met, respectively, using UCSF ChimeraX 1.6.1[45]. The models were partially rebuilt in Coot 0.9.8.1[46].

For the tRNA-Leu structure, tRNA-Ser (PDB: 1SER) was used as a starting model, which was rigid body docked into the cryo-EM density

of SLFN11[wt] bound tRNA-Leu using UCSF ChimeraX 1.6.1[45]. The tRNA model was subjected to interactive molecular-dynamics flexible fitting using ISOLDE 1.6.0[47] and the sequence was mutated to that of tRNA-Leu and partially rebuild in Coot 0.9.8.1[46]. Missing parts were built de-novo. Structures of SLFN11[wt] bound to tRNA-Leu in a pre- and post-cleavage states were processed analogously. The tRNA-Met structure was modeled using similar steps as for tRNA-Leu, with tRNA-fMet (PDB: 7CHD) serving as the starting model.

The atomic models of SLFN11[wt] bound to tRNA-Leu or tRNA-Met were improved using ISOLDE 1.6.0[47] and real space refined in PHENIX 1.20.1-4487[48,49].

The AlphaFold2[50] model of SLFN11 served as the starting model for ATP-bound SLFN11[S753D]. The AlphaFold2 model was rigid body docked in the cryo-EM density of SLFN11[S753D] using UCSF ChimeraX 1.6.1[45]. The model was partially rebuilt in Coot 0.9.8.1[46]. The atomic model was improved by ISOLDE 1.6.0[47] and real space refined in PHE-NIX 1.20.1-4487[48,49].

All structure figures were prepared with UCSF ChimeraX 1.6.1[45].

## Oligonucleotides
Unlabeled tRNA-Leu-TAA, tRNA-Met-CAT, and Cy5 labeled tRNA-Leu Δvar were ordered from GenScript as Custom RNA Oligos. 6-FAM labeled tRNA-Leu-TAA was ordered from Metabion as RNA Custom Oligo. All other oligonucleotides were ordered from Metabion. The list of all used oligonucleotides together with their sequences is provided as Supplementary Table 2.

## Nuclease assay
The nuclease activity of SLFN11 was examined by a gel-based nuclease assay. The nuclease reaction was performed in nuclease buffer (25 mM Tris pH 7.5, 120 mM NaCl, 2 mM MgCl₂, 1 mM DTT) as single-batch reactions. 2 mM MnCl₂ was added if not stated otherwise. SLFN11[wt] was added to the mixture (50 nM, 25 nM, 12.5 nM, or 6.25 nM final concentration) and incubated on ice for 20 min. A final concentration of 25 nM was used for phosphomimetic mutants (SLFN11[S219D], SLFN11[T230D], and SLFN11[S753D]). 200 nM tRNA in nuclease buffer was incubated for 5 min at 95 °C and then gradually cooled down to 20 °C. Reactions were started by adding the substrate (6-FAM labeled tRNA-Leu) with a final concentration of 50 nM and incubated at 37 °C for 60 min. At the indicated time points, 10 ul of the reactions were mixed with 10 ul of loading dye (15% Ficoll, 20 mM Tris pH 7.6, 40 mM NaCl) and boiled at 95 °C for 5 min. Samples were applied to a self-cast 15% denaturing polyacrylamide gel (Rotiphorese® DNA sequencing system). Gels were run in 0.5× TBE at 270 V (Bio-Rad) for 50 min. Gels were imaged using a Typhoon™ FLA 7000 (GE Healthcare) and visualized using GIMP 2.10.28. The fluorescence of uncleaved and cleaved bands was quantified using ImageJ 1.8.0_345[51]. Data are represented as mean values +/− SD from three independent experiments and plotted with Prism 6.07 (GraphPad Software).

The effect of the variable loop on the nuclease activity of SLFN11 was examined using a gel-based nuclease assay as described before. The final concentration of SLFN11[wt] was 25 nM. Both tRNA-Leu (6-FAM) and tRNA-Leu Δvar (Cy5) were added with a final concentration of 25 nM each. Data are represented as mean values +/− SD from three independent experiments and plotted with Prism 6.07 (GraphPad Software).

The analysis of cryo-EM samples was examined using a gel-based nuclease assay as described before. The samples were diluted to a final concentration of 25 nM SLFN11[wt] and the NaCl concentration was adjusted to 100 mM. tRNA (either tRNA-Leu or tRNA-Met) was added to reach a final concentration of 50 nM. Subsequently, MnCl₂ was added to a final concentration of 2 mM and the samples were incubated at room temperature for 10 minutes. The experiment was performed in duplicates.

The analyses of the endonucleolytic cleavage of tRNA-Leu at various SLFN11 concentrations used in MST experiments were

examined using a gel-based nuclease assay as described before. Dilution series of SLFN11[wt], SLFN11[S219D], SLFN11[T230D], and SLFN11[S753D] were performed as in MST experiments. MnCl₂ was added to a final concentration of 1 mM. Subsequently, tRNA-Leu was added to reach a final concentration of 5 nM and the samples were incubated at room temperature for 20 minutes.

Uncropped gels are provided in a Source Data file.

## Microscale thermophoresis (MST)
MST measurements were conducted to determine the affinity of SLFN11 and SLFN11 phosphomimetic mutants to tRNA-Leu or tRNA-Met. A SLFN11 dilution series (2x dilution series from 1800 to 0.055 nM (SLFN11 dimer concentration)) was prepared in MST-buffer (25 mM Tris pH 7.5, 150 mM NaCl, 2 mM MgCl₂, 1 mM DTT). The protein dilutions were mixed with 10 nM 6-FAM labeled tRNA (6-FAM at 5′-end) in MST-buffer, supplemented with 2 mM MnCl₂, in a 1:1 (v/v) ratio (final volume: 20 μl). The samples were incubated at room temperature for 20 min. MST measurements were conducted with a Monolith NT.115 instrument, equipped with the NanoTemper Control 1.1.9 software (NanoTemper Technologies), using standard capillaries. Measurements were performed at 100% LED power and 100% laser power with a laser on time of 10 s at 22.6 °C. All MST experiments were done in triplicates. The data were analyzed using the NanoTemper MO.Affinity Analysis v2.3 software (NanoTemper Technologies) and the *F_norm* values were exported and plotted to Eq. (1) using Prism 6.07 (GraphPad Software).

$$f(c_L) = U + (B - U) * \frac{c_L + c_T + K_d - \sqrt{(c_L + c_T + K_d)^2 - 4 * c_L * c_T}}{2c_T} \quad (1)$$

$U$ = *F_norm* of unbound state, $B$ = *F_norm* of bound state, $c_L$ = concentration of SLFN11, $c_T$ = concentration of tRNA, $K_d$ = dissociation constant

The data were scaled to a range of 0 to 1 by min-max normalization according to Eq. (2).

$$F\_norm\_scaled = \frac{F_{norm} - U}{B - U} \quad (2)$$

As the phosphomimetic mutants did not reach the upper plateau, and thus did not allow the estimation of $B$, $B$ of the wild type data was used for the min-max normalization of the measurements of the phosphomimetic mutants. Finally, the data were plotted to Eq. (1) with a constant value of $c_T$ = 5 nM, $U$ = 0 and $B$ = 1, using Prism 6.07 (GraphPad Software).

Source data are provided in a Source Data file.

## Nano differential scanning fluorimetry (nanoDSF)
Binding of SLFN11 to various substrates was examined by nanoDSF (Tycho NT.6, NanoTemper Technologies). 300 nM SLFN11 was incubated without or with corresponding nucleotides (1 mM) in nanoDSF buffer (25 mM Tris pH 7.5, 60 mM NaCl, 2 mM MgCl₂, 1 mM DTT) for 30 min on ice. The interaction of SLFN11 with DNA was analyzed similarly, where SLFN11 was incubated with 300 nM 50 nt ssDNA or 300 nM 50 bp dsDNA in nanoDSF buffer, respectively. The samples were loaded into glass capillaries and the internal fluorescence at 330 nm and 350 nm was measured while a thermal gradient was applied. Data were analyzed using the internal Tycho NT.6 software 1.3.2.880 and plotted with Prism 6.07 (GraphPad Software). Data are represented as mean values +/− SD from three independent experiments and plotted with Prism 6.07 (GraphPad Software).

Source data are provided in a Source Data file.

## Electrophoretic mobility shift assay (EMSA)

Binding of SLFN11 to nucleic acid substrates was monitored by electrophoretic mobility shift assay (EMSA). 37.5–300 nM SLFN11 was incubated with 40 nM 6-FAM labeled substrates at 4 °C for 30 min in EMSA buffer (25 mM HEPES pH 7.5, 60 mM KCl, 8% glycerol, 2 mM $MgCl_2$, 1 mM DTT). Samples were mixed with loading dye (15% Ficoll, 20 mM Tris pH 7.6, 40 mM NaCl) and applied to a NativePAGE 3-12% Bis-Tris gel (Thermo Fisher). The electrophoresis was performed in 1× NativePAGE running buffer (Thermo Fisher) at 100 V for 120 min at 4 °C. The gels were imaged using a Typhoon™ FLA 7000 (GE Healthcare) and analyzed using GIMP 2.10.28.

Uncropped gels are provided in a Source Data file.

## ATP hydrolysis assay

A fluorescence-based ATPase assay was conducted to determine the ATPase rate of SLFN11[52]. SLFN11$^{wt}$ or SLFN11$^{S753D}$ (200 nM) was incubated with 200 nM of different DNA or RNA substrates in ATPase buffer (25 mM Tris pH 7.5, 50 mM NaCl, 1 mM DTT, 2 mM $MgCl_2$, 0.1 mg ml$^{-1}$ BSA) at 4 °C for 30 min. ΦX174 virion DNA (New England BioLabs) was used at a concentration of 40 ng µl$^{-1}$. RPA was used at a concentration of 400 nM. SLFN11:substrate complexes were combined with 0.1 mM NADH in reaction buffer (25 mM Tris pH 7.5, 50 mM NaCl, 1 mM DTT, 2 mM $MgCl_2$, 0.1 mg ml$^{-1}$ BSA, 0.5 mM PEP, 1 mM ATP, 25 U ml$^{-1}$ lactate dehydrogenase/pyruvate kinase (Sigma-Aldrich)) in a 384 well plate (Greiner). Hexokinase from *Saccharomyces cerevisiae* (2.5 nM, Sigma-Aldrich) supplemented with 300 µM glucose served as positive control. The fluorescence of NADH was measured at 25 °C using an Infinite M1000 microplate photometer (Tecan). The reaction was monitored for 60 min (20 s intervals) using an excitation wavelength of 340 nm and an emission wavelength of 460 nm. Data were analyzed using Prism 6.07 (GraphPad Software).

Source data are provided in a Source Data file.

## Mass photometry

The molecular mass of SLFN11$^{wt}$ and phosphomimetic mutants in solution was determined by mass photometry. All mass photometry measurements were carried out using an OneMP mass photometer (Refeyn). Prior to each measurement the focus was adjusted by applying 19 µl mass photometry buffer (25 mM Tris pH 7.5, 2 mM $MgCl_2$, 1 mM DTT with variable concentrations of NaCl) to a new flow chamber. SLFN11 was diluted in sterile filtered mass photometry buffer to a final concentration of 50 nM, immediately prior to mass photometry measurements. Movies were recorded for 60 s and data were collected and analyzed using Refeyn Acquire$^{MP}$ 2.3 and Refeyn Discover$^{MP}$ 2.3, respectively.

## Reporting summary

Further information on research design is available in the Nature Portfolio Reporting Summary linked to this article.

## Data availability

The data supporting the findings of this study are available from the corresponding authors upon request. The coordinates of the SLFN11$^{wt}$ bound to tRNA-Leu and tRNA-Met structures have been deposited in the Protein Data Bank (PDB) under the accession codes 9ERE and 9ERF, respectively. The cryo-EM reconstructions are available at the Electron Microscopy Data Bank (EMDB) under the EMBD accession codes EMD-19913 and EMD-19914, respectively. The coordinates of the SLFN11$^{wt}$ bound to tRNA-Leu in pre- and post-cleavage state have been deposited in the PDB under the accession code 9GMW and 9GMX. The cryo-EM reconstructions are available at the EMDB under the EMBD accession codes EMD-51456 and EMD-51457, respectively. The coordinates of the SLFN11$^{S753D}$ structure have been deposited in the PDB under the accession code 9ERD and the cryo-EM reconstruction is available at the EMDB under the EMBD accession code EMD-19912. Source data are provided with this paper.

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

## Acknowledgements

We acknowledge Dr. Daniel Bollschweiler from the Max-Planck Institute of Biochemistry for help with the mass photometry experiments and data evaluation. We thank all members of the Hopfner lab for helpful discussions. We acknowledge support by the Deutsche Forschungsgemeinschaft (DFG, German Research Foundation) – Project-ID 210592381 – SFB 1054 (to K.L.), the Gottfried Wilhelm Leibniz-Prize (to K.-P.H.) and the European Research Council (ERC Advanced Grant INO3D, to K.-P.H.).

## Author contributions

M.K., F.J.M., and K.L. conceived the project. M.K., F.J.M., and K.L. designed all structural and biochemical experiments. M.K. and F.J.M. conducted all biochemical experiments. G.W. assisted with MST experiments. M.K. and F.J.M. conducted all structural experiments. F.J.M., M.K., and K.L. carried out the cryo-EM data collection and analysis. K.-P.H. helped with the analysis and interpretation of the results. M.K., F.J.M., and K.L. wrote the manuscript. K.L. and K.-P.H. provided funding. All authors discussed and commented on the results and the manuscript.

## Funding

## Competing interests

The authors declare no competing interests.
