## [Transparent Peer Review file · Nature Communications]

Phosphorylation-mediated conformational change regulates human SLFN11

Corresponding Author: Dr Katja Lammens

Version 0:

Reviewer comments:

Reviewer #1

(Remarks to the Author)

SLFN11 is a multifunctional protein that has been demonstrated to bind and block stalled DNA replication forks, enhancing chemotherapeutic sensitivity, and bind and digest preferentially type II tRNAs, functioning in innate antiviral immunity. Previously, this laboratory solved the structure of wild-type SLFN11 in its dimeric apoenzyme state, with tRNA bound to the RNase domain, or with single-stranded DNA bound to the helicase domain. These studies suggested that SLFN11 dimerization is required for tRNA binding and digestion, and that the helicase domain of wild-type SLFN11 is locked in an autoinhibited state, neither binding nor hydrolyzing ATP.

In this manuscript, the authors report the solution of the Cryo-EM structures of phosphorylation site mutants of SLFN11, with or without bound substrate tRNA. SLFN11 was previously shown to be phosphorylated on S218, T230, and S753. Phosphomimetic mutations of these sites correlated with inhibition of SLFN11 antiviral activity, and dephosphorylation of S753 is required for stalled replication fork blockage, but the biochemical mechanism of these effects was unclear. Here, Kugler et al. solve the cryo-EM structures of these phosphomimetic substitutions of SLFN11.

Unlike wild-type SLFN11, SLFN11 S753D is strictly monomeric and can bind ATP. A large rotation of the helicase domain disrupts one of the two homodimeric contacts, destabilizing the dimeric form, and moves the interdomain helix into an open conformation, permitting ATP binding. This interdomain helix locks the helicase domain of wild-type SLFN11 in an autoinhibitory state unable to bind ATP. Although SLFN11 S753D could bind ATP, there was no evidence of ATP hydrolysis. Moreover, SLFN11 S753D could no longer bind single-stranded DNA.

The authors furthermore reported the structure of SLFN11 with phosphomimetic mutations of the two phosphorylation sites near the nuclease active site, S219 and T230. Like wild-type SLFN11, S219D and T230D exist in a monomer-dimer equilibrium, fail to bind ATP, and are able to bind single-stranded DNA. All three individual phosphomimetic mutations of SLFN11, S219D, T230D, and S753D, decrease SLFN11 tRNase activity. S219 directly contacts bound tRNA, T230 lies in close proximity to the nuclease active site, and S753D disrupts the dimeric structure required for tRNA binding. Interaction of the tRNA variable loop with the nuclease domain may contribute to preference for type II tRNAs over type I.

These are noteworthy results, mechanistically accounting for many SLFN11 functional observations, and comprising a logical next step from the group's previous SLFN11 publication. The structural analysis and biochemical experiments are well done and support the authors' conclusions. The only critiques I have are discussion points.

First, the cryo-EM structure of wild-type SLFN11 with tRNA shows that the tRNA has been cleaved by SLFN11. This observation nicely supports the proposed mechanism of tRNA cleavage, whereby only one protomer of the SLFN11 dimer is competent for cleavage. Please explain why the cleaved tRNA is not released from SLFN11 in the structure.

Second, it seems from these data and the group's previous publication that SLFN11 is not an active helicase, although the manuscript does not actually state this. Please explicitly discuss whether or not SLFN11 is an active helicase and what impact the answer to this question could have on the observed biological functions of SLFN11.

Third, the Discussion section refers to data in the manuscript. It would be helpful to refer to the relevant figures.

Finally, are there any thoughts on why the helicase and RNase activities of SLFN11 are encoded in a single protein? Are there any data to support a role for DNA replication fork blocking in the SLFN11-mediated antiviral response?

Reviewer #2

(Remarks to the Author)

Kugler and Metzner et al. present a functional and structural characterization of the human SLFN11 protein that extends their prior study (Metzner et al., 2023). SLFN11 is a bifunctional enzyme possessing an ATPase and tRNA-nuclease activity. It is known that SLFN11 can sensitize tumor cells to DNA damaging agents and counteract viral infections such as HIV-1 by interfering with translation. Although SLFN11 has been characterized structurally and biochemically before by the authors of this manuscript, it remains poorly understood how SLFN11 is regulated in the cell, that is, how SLFN11 can be activated and deactivated. A current hypothesis is that phosphorylation of SLFN11 might be a key player in this process. Kugler and Metzner et al. introduce the cryo-EM structure of a phosphomimetic mutant of SLFN11 that prevents dimerization and allows for ATP binding. They further characterize various biochemical properties of this mutant in comparison to the WT protein. The authors next provide cryo-EM structures of SLFN11 bound to type I and II tRNAs. At last, they further characterize ssDNA binding and nuclease activity of SLFN11 WT and phosphomimetic mutants. The authors summarize that their study gives new insights in how SLFN11 is regulated by phosphorylation.

The authors describe the structures and biochemical experiments in great depth, however, the overall flow of the text is difficult to follow (see major point #2). Additionally, the interpretation of the nuclease activity assays appears questionable (major point #1) and some very important information is missing with respect to cryo-EM structure model building (major point #3).

The manuscript still provides important advances in the understanding of SLFN11 regulation, however, the experimental and textual issues have to be resolved before publishing.

Major points:

#1 Interpretation of the nuclease activity assay (Figure 4c-d)

The nuclease assay (Figure 4) shows variation of tRNA cleavage over time. The most striking observation here is that, depending on the enzyme concentration, they observe saturation of substrate-product conversion at different percentages of cleaved substrate. In particular, at high enzyme concentrations, the reaction saturates at ~60% of the substrate cleaved, whereas the lowest tested enzyme concentration saturates at ~20%. The authors speculate that this might be caused by product inhibition or misfolded substrate, which would both be valid explanations. However, the authors neglect a third possibility, which is a change in formation of the enzyme(E)-substrate(S) complex when they work at concentrations lower than the K_d of the enzyme. By lowering the enzyme concentration, they would simply reduce the concentration of E-S complex formed and thereby lower the speed of the reaction as it is then limited by the formation of the ES-complex. Once the reaction starts, some substrate will be consumed leading to formation of almost zero ES-complex and hence saturation of the assay in dependence of the initial amount of enzyme. Of course, this cannot be distinguished from product inhibition (which also reduces formation of the E-S complex).

Accordingly, the differences they observe for the phosphomimetic mutants could either be caused by product inhibition (less efficient product release upon mutation) or working at concentrations far away from the K_d (mutations increase K_d [=decrease affinity] and therefore reduce the amount of E-S being formed). As phosphomimetic mutations should introduce a repelling charge for the RNA substrate, the latter case seems more likely (reduced affinity).

Therefore, I think the assay is actually not very conclusive, except giving hints that something is different between wt and the mutants. Furthermore, the authors report that they measure reaction rates (l. 276ff). However, they do not report measurements and calculations for relevant rate parameters for example the K_m . I therefore suggest the authors first measure the binding affinity of the tRNA to a nuclease dead SLFN11 first. If the authors think there is a difference in how fast the enzyme converts the substrate, they could then perform activity assays in a regime where formation of the E-S is not the limiting factor at the beginning of the reaction, which enables measurement of substrate-product conversion over time at different substrate concentrations enabling determination of the K_m .

In summary, the assay results cannot be used for the conclusions made by the authors. They should consider doing affinity measurements or measurements of enzyme parameters like the K_m .

#2 Text flow and readability

Unfortunately, the manuscript does not read very well. While the language is not a problem, the text flow does not help to understand the content. I agree that this is a rather subjective point, however, I am convinced the manuscript can be much improved when carefully edited by the authors. Specifically, the content of the result sections does not always correspond to the headers and the order of how results are presented and interpreted seems sometimes erratic. For example, the section "Overall structure of phosphomimetic SLFN11 S753D" starts as expected with a clear motivation and description of the experiment performed (cryo-EM). This is followed by investigating the multimeric state (mass-photometry), binding of ATP, ADP and ATPgS by melting curves, then a comparison to SLFN5, then going back to describing ATP binding in the structure. There are also quite sudden jumps in the text that are sometimes difficult to follow. Sections seem to discuss too many aspects at once, which are also sometimes unrelated to the section title. Several times, it seems that the order of sentences should be shuffled to make it more logic (for example lines 195-199). My recommendation is to carefully restructure the manuscript and make sure to only follow up one thing at a time and focus on the main advances (which are there for sure!). A clear structure for each result section like motivation-describe experiment-describe result-interpret result-conclude could help.

#3 Maps used for model building

In the methods, the authors do not state which cryo-EM reconstructions were used for model building and model refinement. As the choice of the map has a big impact on the outcome of these steps, the authors must provide this information in the methods. Further, the authors used DeepEMhancer to postprocess the reconstructions. DeepEMhancer was trained on known cryo-EM reconstructions processed with LocScaI, which is a model-based sharpening method. When applied to cryo-EM reconstructions, DeepEMhancer aims to “reproduce” learned protein-like features from noisy data, which is prone to produce artifacts. Were these maps used for model building and refinement? If yes, how do the authors ensure that the deep-learning based map modification did not introduce artifacts that influence model building and or refinement?

#4 Figure 5 and model conclusion

In lines 346ff and Figure 5, the authors put a model forward to explain the regulation of SLFN11 nuclease activity by phosphorylation. While the DDA related function is proposed to be tRNA cleavage in the nucleus, why would tRNA cleavage be only happening in the cytoplasm in case of the ssDNA response (Figure 5B)? This seems inconsistent and should be clarified in the discussion and the model figure. Additionally, Fig 5B implies that ssDNA binding induces dimerization. However, the authors show that dimerization is mainly driven by the phosphorylation state. I guess this is a misunderstanding, but this could be resolved by making clearer in the model figure that dimerization and ssDNA binding are separate events.

Other points:

It did not get clear from the manuscript what the role of ATP binding (and potentially hydrolysis) is for SLFN11. Could the authors elaborate on this in the discussion? Could ATP binding/hydrolysis play a role in the function or regulation of SLFN11?

The authors claim in the discussion that they provide an explanation how SLFN11 does distinguish type I and II tRNAs. However, there is no such evidence in the text (or its not explained well). How does SLFN11 achieve this? Especially the variable loop has not been investigated further. Could the authors measure affinities of the same tRNA +/- the variable loop to actually highlight its importance?

Many of the points that follow are actually related to major point #2:

Line 30: PARP was not introduced as abbreviation before

Line 36: RPA was not introduced as abbreviation before

Introduction: the authors might want to introduce the phosphatases mentioned in lines 341-345 already here as it is important prior knowledge when reading the manuscript. Also, it should be more highlighted that the kinases phosphorylating SLFN11 are unknown.

Line 73-77: The authors might consider to introduce the domain architecture in the introduction.

Line 90: The authors suddenly mention an ATP-free reconstruction. Where does this come from? It was not introduced. Please make sure you introduce the experiments before you present the result.

Line 109: The authors refer to the structure of a SLFN11 E209A mutant. Where does this come from? Please reference the structure with PDB ID and literature.

Line 120 and throughout: There are several occurrences of “inflection temperature”. Please consider to change them to “melting temperature” as “inflection” is not very intuitive (albeit somewhat correct).

Line 123: The authors refer to SLFN5 here, which seems a bit random. How is that important? Please make clear why you compare to SLFN5 here (and also in the discussion).

Line 125-127: The authors use somewhat reverse logic: their preparation method is in line with their observation in the cryo-EM (as if they first did the EM and then the preparation). Consider to inverse the sentence. Further, the presence of density for ATP does not only “suggest” an ATP bound state (unless the authors are not sure about it), but rather “shows” that ATP is bound.

Line 132: Consider to use “change” instead of “switch” throughout as switch can have two meanings that could be confused.

Line 137 and 138: Figure 2B is referenced before 2A. Consider to swap the panels.

Line 139: The change that is observed in the ID region is quite substantial and might be better described as “refolding” and not just as a conformational change.

Line 140-142: The sentence about SLFN5 seems again randomly placed here. What is the “related conformational change”? How is important to relate to SLFN5 here? If the conformational change is important, describe it here and do a comparison.

Line 147: The linker domain is mentioned with Figure 2 referenced. Please also highlight the linker domain in Fig 2.

Line 150: When mentioning ID region II, the region should be labeled in the relevant figures (like in Fig 2A).

Line 151: When the authors speculate that the ID region keeps the C-lobe of the helicase domain in place, they should refer to the SLFN11 E209A structure. Otherwise it is hard to understand where the idea comes from.

Line 171: Add "mutation" behind "S753D"

Line 172: Which conformational change and how is it hindered? Which loop is meant? Please clarify.

Line 195 and throughout: Please use a consistent format for the tRNA names (super script or not, but not both).

Line 195-197: How do these structures compare to your previous structure of SLFN11?

Line 205: The authors speculate about direct sequence readout by Q35. Have the authors looked at the sequences of tRNAs targeted by SLFN11 if they carry a specific base at that position?

Line 209: The authors speculate, that the variable loop in tRNA determines the binding affinity of tRNAs to SLFN11. The authors might want to actually test this.

Line 216ff: Please show densities for the cleaved and uncleaved tRNA in the supplement and reference them here.

Line 220: The authors report on the presence/absence of metal ions in the structure. Please show densities for the metal ions in the supplement to support these statements.

Line 222: The authors conclude that the metal in the structure must be Mn²⁺ because it was present in the sample preparation. However, Mg²⁺ was present at the same concentration during sample preparation. Although I agree that it is likely Mn²⁺, the argument they use here is really weak. The authors should clarify this. Is the enzyme maybe more active with Mn²⁺ vs Mg²⁺, which would indicate higher affinity for Mn²⁺?

Line 226ff: The authors report on slight differences in the way the tRNAs are bound. However, they do not further elaborate what the consequence of this is. Would this be a mechanism for specificity? Would the affinity be different between the two modes?

Line 269: The authors report that SLFN11 mutants bind ssDNA but not dsDNA. Please elaborate what the consequence of this observation is.

Line 281 and 288: The authors report a "significant reduction", however, do not provide numbers or statistical tests to support this. Please also see major point #2

Line 389: As T230 is specific for SLFN11, the site is indeed unique (not just might be unique).

Line 463 and throughout: Please indicate which mask was used to calculate the FSC (presumably cryoSPARC automasking?).

Line 481: "subject" should be "subjected" (also other occurrences thereafter).

Figure 1a+b: Please also highlight the location of S219 and T230 in the structure.

Figure 3: Consider to show structures instead of the reconstructions.

Figure 3B: label where the active sites are (Nuclease active site I and II)

Figure 3B/C: highlight where C is in B.

Figure 5+6: This figure could benefit from domain labels (in addition to the color code).

Figure 7C: The legend reports that differences are indicated. However, it does not become clear from panel C what are the differences. Maybe use the same color in different shades for the same structure region to make the comparison more intuitive.

Figure 8: The figure contains data that was not mentioned in the text (many different ATPase assays). If the data is relevant, include and describe it in the text. Remove it from the figures otherwise.

Figure 9: It is not quite clear how processing was done here. Also the method text does not help to understand it. Please clarify this.

-- Christian Dienemann (signed review)

Reviewer #3

(Remarks to the Author)

In this work, the authors reveal novel insight into the regulation of the SLFN11 functions and enzymatic activities. They provide cryoEM structures of the SLFN11 phosphomimetic mutant S753D bound to ATP, and of the SLFN11 WT protein bound either to tRNAMet or tRNALeu.

Globally, the study shows that the phosphorylation of S753 prevents the dimerization of SFLN11 and allows the ATP binding. The cryoEM structures with two tRNAs, one with a long variable loop (tRNALeu) and one with a short-one (tRNAMet) give insights into tRNA recognition and cleavage, as well as its regulation by phosphorylation at S219 and T230. Although previous studies have already provided information about the structure of SLFN11 and its interaction with tRNAs, the impact of phosphorylation remains elusive. The manuscript is written in a very clear way. I recommend publication of this article after correction, which would be of hight interest for the broad readership of this journal.

I have several suggestions/comments:

- A general remark : it is difficult to judge of the quality of the map because we have no figure with the model fitted in the map. For instance, it could be informative to see the density for the ions, for the active sites... Figures with density should be added either in the main text or in supplementary information.

- For structures with tRNA, it is difficult to appreciate in the figures what part of the tRNA is directly bound to the protein. It could be informative to have a figure where we can see the L-shaped tRNA and which arms make contacts with SLFN11. Add the name of the arms on the figures and precise which part of the variable loop is visible. I guess not all the variable loop is observed for tRNALeu.

- It is not clear for me why tRNALeu which is a good substrate for SLFN11 is not released after cleavage, so why it is still bound to SLFN11. Could the author provide a gel made in the same condition than the cryoEM sample to be sure that the structure SLFN11/tRNALeu is a post-cleavage structure.

-Why one site of SLFN11 is cleavage-deficient?

- Were the NanoDSF experiments repeated three times? It would be interesting to have the Tm values with SD.

- There is no information about the sequences and production/purification of tRNA samples. This information should be added.

- The SLFN11 protein is produced in insect cells. Have the authors checked the phosphorylation status of the protein after purification?

- Have you tried to model the structure of SLFN11 S219 and T230D using AlphaFold3? Would this help your discussion?

Version 1:

Reviewer comments:

Reviewer #1

(Remarks to the Author)

The authors have successfully addressed my concerns. This manuscript is suitable for publication in Nature Communications.

Reviewer #2

(Remarks to the Author)

The authors have addressed all points raised.

-- Christian Dienemann (signed review)

Reviewer #3

(Remarks to the Author)

The authors have responded to all my points and corrected their paper as requested. I recommend publication of this paper in its revised version.

We thank the editor and reviewers for their positive feedback, as well as their constructive and insightful comments. We have addressed all of the reviewers' points and revised the manuscript accordingly.

In summary, we have made the following changes:

- We restructured and reorganized the manuscript to improve readability and flow of the text.
- Further, we incorporated recent publications (Boon et al., 2024; Zhang et al., 2024) to put our findings into an up to date context.
- To better understand tRNA recognition and cleavage, we collected a new cryo-EM dataset and resolved the structure of SLFN11 bound to tRNA-Leu-TAA in both the pre- and post-cleavage states.
- To clarify the nuclease activity results, we performed Microscale Thermophoresis (MST) assays to calculate the binding constants of SLFN11 to both type I (Met) and type II (Leu) tRNAs. Similarly, we used MST to obtain the binding constants of respective SLFN11 phosphomimetic mutants (S219D, T230D, S753D) to tRNA-Leu. To investigate the importance of the variable arm, we performed competition nuclease assays with tRNA-Leu with or without the variable arm.
- We addressed the raised questions related to the cryo-EM structure model building and map usage.

The reviewers also requested a more detailed and speculative discussion of our data in light of existing literature. In response, we have expanded the discussion section in the revised manuscript. We updated our model for the activation of SLFN11 by DDAs and nuclear immune stimulatory ssDNA, integrating the latest discoveries.

Overall, we believe the manuscript has been substantially improved, and we would like to thank the reviewers for their valuable input, which has led to meaningful adjustments in the manuscript.

Point-by-point response to referees:

REVIEWER COMMENTS

Reviewer #1 (Remarks to the Author):

SLFN11 is a multifunctional protein that has been demonstrated to bind and block stalled DNA replication forks, enhancing chemotherapeutic sensitivity, and bind and digest preferentially type II tRNAs, functioning in innate antiviral immunity. Previously, this laboratory solved the structure of wild-type SLFN11 in its dimeric apoenzyme state, with tRNA bound to the RNase domain, or with single-stranded DNA bound to the helicase domain. These studies suggested that SLFN11 dimerization is required for tRNA binding and digestion, and that the helicase domain of wild-type SLFN11 is locked in an autoinhibited state, neither binding nor hydrolyzing ATP.

In this manuscript, the authors report the solution of the Cryo-EM structures of phosphorylation site mutants of SLFN11, with or without bound substrate tRNA. SLFN11 was previously shown to be phosphorylated on S218, T230, and S753. Phosphomimetic mutations of these sites correlated with inhibition of SLFN11 antiviral activity, and dephosphorylation of S753 is required for stalled replication fork blockage, but the biochemical mechanism of these effects was unclear. Here, Kugler et al. solve the cryo-EM structures of these phosphomimetic substitutions of SLFN11.

Unlike wild-type SLFN11, SLFN11 S753D is strictly monomeric and can bind ATP. A large rotation of the helicase domain disrupts one of the two homodimeric contacts, destabilizing the dimeric form, and moves the interdomain helix into an open conformation, permitting ATP binding. This interdomain helix locks the helicase domain of wild-type SLFN11 in an autoinhibitory state unable to bind ATP. Although SLFN11 S753D could bind ATP, there was no evidence of ATP hydrolysis. Moreover, SLFN11 S753D could no longer bind single-stranded DNA.

The authors furthermore reported the structure of SLFN11 with phosphomimetic mutations of the two phosphorylation sites near the nuclease active site, S219 and T230. Like wild-type SLFN11, S219D and T230D exist in a monomer-dimer equilibrium, fail to bind ATP, and are able to bind single-stranded DNA. All three individual phosphomimetic mutations of SLFN11, S219D, T230D, and S753D, decrease SLFN11 tRNase activity. S219 directly contacts bound tRNA, T230 lies in close proximity to the nuclease active site, and S753D disrupts the dimeric structure required for tRNA binding. Interaction of the tRNA variable loop with the nuclease domain may contribute to preference for type II tRNAs over type I.

These are noteworthy results, mechanistically accounting for many SLFN11 functional observations, and comprising a logical next step from the group's previous SLFN11 publication. The structural analysis and biochemical experiments are well done and support the authors' conclusions. The only critiques I have are discussion points.

We thank the reviewer for the kind appreciation of our work and we hope that we address all issues in the revised version of the manuscript.

First, the cryo-EM structure of wild-type SLFN11 with tRNA shows that the tRNA has been cleaved by SLFN11. This observation nicely supports the proposed mechanism of tRNA cleavage, whereby only one protomer of the SLFN11 dimer is competent for cleavage. Please explain why the cleaved tRNA is not released from SLFN11 in the structure.

Based on our previous results, we do not have a clear explanation for why tRNA is not released from SLFN11 in the structure. However, we addressed this question through additional experiments and provided an explanation based on the obtained results. MST and nuclease cleavage assays suggest that SLFN11 has a slightly higher affinity for cleaved tRNA compared to uncleaved tRNA. Unfortunately, attempts to use tRNA with a supposedly uncleavable phosphorothioated bond at the SLFN11 cleavage site were unsuccessful, as some degree of cleavage of the tRNA was still observed. As a result, we were unable to determine conclusively which form of tRNA binds more effectively.

Also, we collected a new cryo-EM dataset and resolved SLFN11 bound to tRNA-Leu-TAA in the pre- and post-cleavage state. In both cases, tRNA is bound to SLFN11 in a relatively similar conformation. The observation of predominantly cleaved tRNA in our reconstructions further support that SLFN11 could have a higher affinity for cleaved tRNA than for uncleaved tRNA.

Ultimately, the *in vitro* conditions used, differ from those *in vivo*, where tRNAs are modified, and a potential unknown interaction partner—absent in the *in vitro* reaction—may assist SLFN11 in releasing the cleaved tRNA.

Second, it seems from these data and the group's previous publication that SLFN11 is not an active helicase, although the manuscript does not actually state this. Please explicitly discuss whether or not SLFN11 is an active helicase and what impact the answer to this question could have on the observed biological functions of SLFN11.

We added a section explicitly discussing ATPase and helicase activity in the Discussion (line 443 and onward). In particular, we included the following sentences in the manuscript: "Nevertheless, the inability of SLFN11 to hydrolyse ATP independently, combined with the absence of double-stranded DNA-binding elements corresponding to SF1A helicase domains 1B and 2B, demonstrates that the protein alone is not a strand-opening helicase. Therefore, an additional factor, such as a binding partner, modification, or signal, is required to activate its ATPase and helicase activity."

Third, the Discussion section refers to data in the manuscript. It would be helpful to refer to the relevant figures.

We added references to the relevant figures in the discussion section.

Finally, are there any thoughts on why the helicase and RNase activities of SLFN11 are encoded in a single protein? Are there any data to support a role for DNA replication fork blocking in the SLFN11-mediated antiviral response?

In the light of the recent publication (Zhang et al., 2024), it seems likely that the helicase domain of SLFN11 functions as a sensor for CGT-motif containing ssDNA, allowing its translocation to the cytosol, where tRNA cleavage takes place. SLFN11 might be able to detect ssDNA from different endogenous (e.g. DNA damage, stalled replication fork) or exogenous sources (e.g. viruses). Detection of CGT-motif containing ssDNA fragments by SLFN11, might, regardless of their origin, result in translocation of SLFN11 to the cytosol, where tRNA cleavage takes place, eventually leading to ribosome stalling and cell death.

Reviewer #2 (Remarks to the Author):

Kugler and Metzner et al. present a functional and structural characterization of the human SLFN11 protein that extends their prior study (Metzner et al., 2023). SLFN11 is a bifunctional enzyme possessing an ATPase and tRNA-nuclease activity. It is known that SLFN11 can sensitize tumor cells to DNA damaging agents and counteract viral infections such as HIV-1 by interfering with translation. Although SLFN11 has been characterized structurally and biochemically before by the authors of this manuscript, it remains poorly understood how SLFN11 is regulated in the cell, that is, how SLFN11 can be activated and deactivated. A current hypothesis is that phosphorylation of SLFN11 might be a key player in this process. Kugler and Metzner et al. introduce the cryo-EM structure of a phosphomimetic mutant of SLFN11 that prevents dimerization and allows for ATP binding. They further characterize various biochemical properties of this mutant in comparison to the WT protein. The authors next provide cryo-EM structures of SLFN11 bound to type I and II tRNAs. At last, they further characterize ssDNA binding and nuclease activity of SLFN11 WT and phosphomimetic mutants. The authors summarize that their study gives new insights in how SLFN11 is regulated by phosphorylation.

The authors describe the structures and biochemical experiments in great depth, however, the overall flow of the text is difficult to follow (see major point #2). Additionally, the interpretation of the nuclease activity assays appears questionable (major point #1) and some very important information is missing with respect to cryo-EM structure model building (major point #3).

The manuscript still provides important advances in the understanding of SLFN11 regulation, however, the experimental and textual issues have to be resolved before publishing.

We thank the reviewer for his time, and the insightful comments, and hope to address all questions raised in the revised version of the manuscript. We apologize for any difficulty in following the initial version in certain sections. We believe that we have improved the readability and flow of the manuscript. Additionally, we conducted SLFN11-tRNA binding experiments using MST to better interpret the results from the endonuclease cleavage assays. We also performed competition nuclease assays with tRNA-Leu, with or without the variable arm, to investigate the importance of the variable arm. Finally, we clarified which maps were used for cryo-EM structure model building. Overall, we believe we have significantly improved the manuscript in all three major aspects, along with making minor corrections that further enhanced its quality.

Major points:

#1 Interpretation of the nuclease activity assay (Figure 4c+d)

The nuclease assay (Figure 4) shows variation of tRNA cleavage over time. The most striking observation here is that, depending on the enzyme concentration, they observe saturation of substrate-product conversion at different percentages of cleaved substrate. In particular, at high enzyme concentrations, the reaction saturates at ~60% of the substrate cleaved, whereas the lowest tested enzyme concentration saturates at ~20%. The authors speculate that this might be caused by product inhibition or misfolded substrate, which would both be valid explanations. However, the authors neglect a third possibility, which is a change in formation of the enzyme(E)-substrate(S) complex when they work at concentrations lower than the K_d of the enzyme. By lowering the enzyme concentration, they would simply reduce the concentration of E-S complex formed and thereby lower the speed of the reaction as it is then limited by the formation of the ES-complex. Once the reaction starts, some substrate will be consumed leading to formation of almost zero ES-complex and hence

saturation of the assay in dependence of the initial amount of enzyme. Of course, this cannot be distinguished from product inhibition (which also reduces formation of the E-S complex).

Accordingly, the differences they observe for the phosphomimetic mutants could either be caused by product inhibition (less efficient product release upon mutation) or working at concentrations far away from the K_d (mutations increase K_d [=decrease affinity] and therefore reduce the amount of E-S being formed). As phosphomimetic mutations should introduce a repelling charge for the RNA substrate, the latter case seems more likely (reduced affinity).

Therefore, I think the assay is actually not very conclusive, except giving hints that something is different between wt and the mutants. Furthermore, the authors report that they measure reaction rates (l. 276ff). However, they do not report measurements and calculations for relevant rate parameters for example the K_m . I therefore suggest the authors first measure the binding affinity of the tRNA to a nuclease dead SLFN11 first. If the authors think there is a difference in how fast the enzyme converts the substrate, they could then perform activity assays in a regime where formation of the E-S is not the limiting factor at the beginning of the reaction, which enables measurement of substrate-product conversion over time at different substrate concentrations enabling determination of the K_m .

In summary, the assay results cannot be used for the conclusions made by the authors. They should consider doing affinity measurements or measurements of enzyme parameters like the K_m .

We appreciate the detailed analysis of the nuclease activity assays and welcomed the suggestions on how to interpret the obtained results more clearly. As recommended we performed affinity measurements using MST to determine the binding of SLFN11^{wt} and different phosphomimetic mutants to different. Under the conditions tested, and considering the assay's margin of error, SLFN11^{wt} binds to both type I (Met) and type II (Leu) with comparable affinities (Fig. 2E). This is in line with the observed cleavage pattern for these tRNAs in the nuclease cleavage assays (Fig. 2C). We also tested binding of phosphomimetic mutants to tRNA-Leu-TAA by MST (Fig. 3C). The observed binding constants followed the trend of the nuclease cleavage assay analysis of phosphomimetic mutants (Fig. 3D). However, in all assays, binding and cleavage could not be separated, as both uncleaved and cleaved tRNAs were present in the sample under the MST assay conditions (Supplementary Fig. 9). Furthermore, MST measurements using nuclease dead SLFN11 mutants or omitting manganese were not possible, as the affinity to tRNA was not high enough to obtain binding curves (data not shown). Thus, the MST measurements were conducted with SLFN11^{wt} and in the presence of manganese, making it impossible to determine whether cleaved or intact tRNA binds more effectively, and complicating efforts to prove product inhibition.

Unfortunately, our attempts to use tRNA with an uncleavable phosphorothioated bond at the SLFN11 cleavage site were unsuccessful because even this variant was cleaved to some extent. As a result, we were unable to determine exactly which form—cleaved or intact—binds more effectively.

#2 Text flow and readability

Unfortunately, the manuscript does not read very well. While the language is not a problem, the text flow does not help to understand the content. I agree that this is a rather subjective point, however, I am convinced the manuscript can be much improved when carefully edited by the authors. Specifically, the content of the result sections does not always correspond to the headers and the order of how results are presented and interpreted seems sometimes erratic. For example, the section

“Overall structure of phosphomimetic SLFN11 S753D” starts as expected with a clear motivation and description of the experiment performed (cryo-EM). This is followed by investigating the multimeric state (mass-photometry), binding of ATP, ADP and ATPγS by melting curves, then a comparison to SLFN5, then going back to describing ATP binding in the structure. There are also quite sudden jumps in the text that are sometimes difficult to follow. Sections seem to discuss too many aspects at once, which are also sometimes unrelated to the section title. Several times, it seems that the order of sentences should be shuffled to make it more logic (for example lines 195-199). My recommendation is to carefully restructure the manuscript and make sure to only follow up one thing at a time and focus on the main advances (which are there for sure!). A clear structure for each result section like motivation-describe experiment-describe result-interpret result-conclude could help.

We apologize for the difficulty in following certain sections of our manuscript. We have restructured and reorganized it to present one concept at a time more clearly. Additionally, in light of recent discoveries (Boon et al., 2024; Zhang et al., 2024), we have reinterpreted our findings to provide a new perspective. Overall, we believe we have improved the manuscript's readability and flow while also discussing our results within the context of current knowledge in the field.

#3 Maps used for model building

In the methods, the authors do not state which cryo-EM reconstructions were used for model building and model refinement. As the choice of the map has a big impact on the outcome of these steps, the authors must provide this information in the methods. Further, the authors used DeepEMhancer to postprocess the reconstructions. DeepEMhancer was trained on known cryo-EM reconstructions processed with LocScal, which is a model-based sharpening method. When applied to cryo-EM reconstructions, DeepEMhancer aims to “reproduce” learned protein-like features from noisy data, which is prone to produce artifacts. Were these maps used for model building and refinement? If yes, how do the authors ensure that the deep-learning based map modification did not introduce artifacts that influence model building and or refinement?

We apologize for not specifying this information in the Methods section. We added the description which maps were used for model building and refinement in the method section. Additionally, we clarified that the maps post-processed by DeepEMhancer were not used for model building or refinement, but solely for the preparation of the respective figures.

#4 Figure 5 and model conclusion

In lines 346ff and Figure 5, the authors put a model forward to explain the regulation of SLFN11 nuclease activity by phosphorylation. While the DDA related function is proposed to be tRNA cleavage in the nucleus, why would tRNA cleavage be only happening in the cytoplasm in case of the ssDNA response (Figure 5B)? This seems inconsistent and should be clarified in the discussion and the model figure. Additionally, Fig 5B implies that ssDNA binding induces dimerization. However, the authors show that dimerization is mainly driven by the phosphorylation state. I guess this is a misunderstanding, but this could be resolved by making clearer in the model figure that dimerization and ssDNA binding are separate events.

We updated the overall model figure (now Figure 6) in the light of the recent discoveries (Boon et al., 2024; Zhang et al., 2024). We agree that tRNA cleavage could happen in both cytoplasm and also in

the nucleus. Thus, we updated the model accordingly. Furthermore, we visualize SLFN11 dimerization and ssDNA binding as two sequential steps.

Other points:

It did not get clear from the manuscript what the role of ATP binding (and potentially hydrolysis) is for SLFN11. Could the authors elaborate on this in the discussion? Could ATP binding/hydrolysis play a role in the function or regulation of SLFN11?

We added a section explicitly discussing ATPase and helicase activity in the Discussion (line 443 and onward). So far, we do not understand the exact role of ATP binding in SLFN11. ATP binding could stabilize the SLFN11^{S753D} state, which is unable to bind to ssDNA and might thus be an inactive resting state. On the other hand, it was shown that a SLFN11 Walker B mutant is not able to irreversibly block stalled replication forks, despite its recruitment to chromatin. Thus, SLFN11 might undergo a conformational change from the autoinhibited dimer state that is ssDNA binding proficient, to the monomeric and ATP binding proficient state in order for its ATPase to be active and to irreversibly block stalled replication forks.

The authors claim in the discussion that they provide an explanation how SLFN11 does distinguish type I and II tRNAs. However, there is no such evidence in the text (or its not explained well). How does SLFN11 achieve this? Especially the variable loop has not been investigated further. Could the authors measure affinities of the same tRNA +/- the variable loop to actually highlight its importance?

We extended the text and described the specific interactions and the effect of the variable arm from line 167 onward. We also pointed out the structural differences between type I (tRNA-Met) and type II (tRNA-Leu) in Figure 2. Additionally, we performed MST assays to calculate the binding constants of SLFN11 to both types of tRNA. Under the conditions tested, and considering the assay's margin of error, SLFN11 binds to both tRNAs with comparable affinities. To further investigate the importance of the variable arm, we performed competition tRNA-cleavage nuclease assay between tRNA-Leu (FAM-labelled) and tRNA-Leu lacking variable arm (tRNA-Leu Δ var, Cy5-labelled). We observed a strong reduction in tRNA cleavage activity upon deletion of the variable arm, highlighting the importance of this interaction.

Many of the points that follow are actually related to major point #2:

Line 30: PARP was not introduced as abbreviation before

The description of the abbreviation was added to the text.

Line 36: RPA was not introduced as abbreviation before

The description of the abbreviation was added to the text.

Introduction: the authors might want to introduce the phosphatases mentioned in lines 341-345 already here as it is important prior knowledge when reading the manuscript. Also, it should be more highlighted that the kinases phosphorylating SLFN11 are unknown.

We introduce the phosphatases in the introduction and highlight that the kinases of SLFN11 are unknown in the introduction.

Line 73-77: The authors might consider to introduce the domain architecture in the introduction.

We change the text accordingly that we introduce the domain architecture in the introduction.

Line 90: The authors suddenly mention an ATP-free reconstruction. Where does this come from? It was not introduced. Please make sure you introduce the experiments before you present the result.

We added the text in the results section describing the origin of the ATP free reconstruction. The ATP-free reconstruction was one of the obtained classes after heterogenous refinement, where helicase domain C-lobe was not resolved. We updated the corresponding method section and labelled the corresponding classes (ATP-free and ATP-bound) in cryo-EM data analysis of SLFN11^{S753D} (Supplementary Fig. 13).

Line 109: The authors refer to the structure of a SLFN11 E209A mutant. Where does this come from? Please reference the structure with PDB ID and literature.

We added the corresponding EMDB accession code and citation to this reconstruction.

Line 120 and throughout: There are several occurrences of “inflection temperature”. Please consider to change them to “melting temperature” as “inflection” is not very intuitive (albeit somewhat correct).

We understand the point of the reviewer as “melting temperature” is more intuitive than “inflection temperature”. Yet, we decided to use the term “inflection temperatures” as it is describing better the process happening in Tycho NT.6. The melting point or melting temperature (T_m) by definition describes reversible denaturation processes which are in equilibrium throughout an experiment, typically requiring slow heating rates. These criteria are rarely met for thermal unfolding of proteins, which is typically an irreversible process. In the Tycho NT.6 with its fast heating rate, inflection points therefore do not correspond to the physical melting point of the protein, and are simply called inflection temperatures (T_i) instead.

Line 123: The authors refer to SLFN5 here, which seems a bit random. How is that important? Please make clear why you compare to SLFN5 here (and also in the discussion).

We moved the comparison of SLFN11 with SLFN5 from the results part to the discussion. We compare our results to SLFN5, another subgroup III Slfn family member, as its conformation of the helicase domain and some biochemical properties (e.g. nucleotide binding) resemble the properties of SLFN11^{S753D}.

Line 125-127: The authors use somewhat reverse logic: their preparation method is in line with their observation in the cryo-EM (as if they first did the EM and then the preparation). Consider to inverse the sentence. Further, the presence of density for ATP does not only “suggest” an ATP bound state (unless the authors are not sure about it), but rather “shows” that ATP is bound.

We reformulate the sentence to improve its clarity and meaning.

Line 132: Consider to use “change” instead of “switch” throughout as switch can have two meanings that could be confused.

We changed “switch” to “change” in the corresponding places in the manuscript.

Line 137 and 138: Figure 2B is referenced before 2A. Consider to swap the panels.

We swapped the panels of Figure 2 (now Figure 5) accordingly.

Line 139: The change that is observed in the ID region is quite substantial and might be better described as “refolding” and not just as a conformational change.

We thank the reviewer for the suggestion. Even though the conformational change is quite substantial, we would like to keep the term “conformational change”, as we think it is more suitable for the description of the observed effect.

Line 140-142: The sentence about SLFN5 seems again randomly placed here. What is the “related conformational change”? How is important to relate to SLFN5 here? If the conformational change is important, describe it here and do a comparison.

We moved the comparison of SLFN5 from the results part to the discussion. The “related conformational change” is referring to the rotation of the helicase domain previously observed between SLFN11^{wt} and SLFN5^{wt} (Metzner et al., 2021; Metzner et al., 2022). As the conformations between the helicase domains of SLFN5^{wt} and SLFN11^{S753D} are very similar, we observe the same conformation change between SLFN11^{wt} and SLFN11^{S753D} as we previously observed between SLFN11^{wt} and SLFN5^{wt}. The side by side comparison of SLFN11^{S753D} and SLFN5^{wt} is presented in Supplementary Fig. 18.

Line 147: The linker domain is mentioned with Figure 2 referenced. Please also highlight the linker domain in Fig 2.

We highlighted the linker domain in Figure 2 (now Figure 5).

Line 150: When mentioning ID region II, the region should be labeled in the relevant figures (like in Fig 2A).

We labelled the ID region II in Figure 2A (now Figure 5B).

Line 151: When the authors speculate that the ID region keeps the C-lobe of the helicase domain in place, they should refer to the SLFN11 E209A structure. Otherwise it is hard to understand where the idea comes from.

Now we include the comparison with SLFN11^{E209A} in the text in lines 366 to 370.

Line 171: Add “mutation” behind “S753D”

We added “mutation” behind “S753D”.

Line 172: Which conformational change and how is it hindered? Which loop is meant? Please clarify.

We rewrote the text in more detail to properly describe the observed structural changes. We used the exact protein residue boundaries to describe the structural motives. Also, we added these residue boundaries as the labels in corresponding Supplementary Fig. 7 (now Supplementary Fig. 17).

Line 195 and throughout: Please use a consistent format for the tRNA names (super script or not, but not both).

We unified the format for the tRNA names throughout the manuscript.

Line 195-197: How do these structures compare to your previous structure of SLFN11?

We compared the structures of SLFN11 bound to tRNA-Leu and tRNA-Met with the previously published structure of SLFN11^{wt} (PDB: 7ZEL; Metzner et al., 2022) and added the following sentence to the manuscript: “Compared to the SLFN11^{wt} apoenzyme (PDB: 7ZEL) we observe only minor conformational changes of amino acid side chains in close proximity to nuclease active site I.”

Line 205: The authors speculate about direct sequence readout by Q35. Have the authors looked at the sequences of tRNAs targeted by SLFN11 if they carry a specific base at that position?

We aligned all human Leu and Ser type II tRNAs in Supplementary Fig. 7D. Of all Leu tRNAs, only tRNA-Leu-TAAs have guanosine at the position that interacts with Q35 of SLFN11. Thus, this base might help to distinguish between the individual variants of tRNA-Leu. Yet, as this base is not unique for tRNA-Leu-TAA, other interactions might also play a role. An AlphaFold model of the tRNA-Leu-SLFN11 complex, extending beyond the density-based model, suggests that SLFN11 residues E225 and R229 provide additional sequence readout of several bases in the tRNA acceptor stem (Supplementary Fig. 7C). Since most sequence differences between tRNA-Leu and tRNA-Ser are within the variable arm and the acceptor stem, these contacts could also play a role in specific tRNA recognition.

Line 209: The authors speculate, that the variable loop in tRNA determines the binding affinity of tRNAs to SLFN11. The authors might want to actually test this.

To investigate the importance of the variable arm, we performed a competitive tRNA cleavage assay with tRNA-Leu (FAM-labelled) and tRNA-Leu lacking the variable arm (tRNA-Leu Δ var, Cy5-labelled). We observed a strong reduction in tRNA cleavage activity upon deletion of the variable arm, highlighting the importance of this interaction for tRNA cleavage.

Line 216ff: Please show densities for the cleaved and uncleaved tRNA in the supplement and reference them here.

We added figures of the cryo-EM densities of nuclease active sites I and II for the SLFN11 complexes with tRNA-Leu (Supplementary Fig. 4B) and tRNA-Met (Fig. 2B). Furthermore, we resolved the cleaved and uncleaved tRNA-Leu states in a newly collected cryo-EM dataset. We show the densities from this new dataset for the uncleaved (pre-cleavage) and cleaved (post-cleavage) tRNA-Leu in Supplementary Fig. 6A.

Line 220: The authors report on the presence/absence of metal ions in the structure. Please show densities for the metal ions in the supplement to support these statements.

We show the densities for metal ions for all four structures we obtained: tRNA-Met is shown in Fig. 2B, tRNA-Leu in Supplementary Fig. 4B, and tRNA-Leu in pre- and post-cleavage state in Supplementary Fig. 6A.

Line 222: The authors conclude that the metal in the structure must be Mn²⁺ because it was present in the sample preparation. However, Mg²⁺ was present at the same concentration during sample preparation. Although I agree that it is likely Mn²⁺, the argument they use here is really weak. The authors should clarify this. Is the enzyme maybe more active with Mn²⁺ vs Mg²⁺, which would indicate higher affinity for Mn²⁺?

We previously observed endoribonuclease activity of SLFN11 only in the presence of Mn²⁺ but not in the presence of Mg²⁺ (Metzner et al., 2022). We reformulated the sentence and added the reference to the corresponding paper into our manuscript.

Line 226ff: The authors report on slight differences in the way the tRNAs are bound. However, they do not further elaborate what the consequence of this is. Would this be a mechanism for specificity? Would the affinity be different between the two modes?

We extended the discussion about the observed differences in conformations between tRNA-Leu and tRNA-Met bound to SLFN11.

The conformation of tRNA-Leu and tRNA-Met is almost identical in the cleavage-deficient site II, though it slightly differs in nuclease active site I, including a slightly changed positioning of the secondary manganese ion (Supplementary Fig. 4C). As nuclease active site I is the cleavage-proficient site, the observed differences between tRNA-Leu and tRNA-Met might contribute to the cleavage specificity. A more striking difference between the two structures is the presence of an additional

interaction site for the variable loop with the nuclease domain in the SLFN11 structure bound to tRNA-Leu. This interaction results in a tilting of tRNA-Leu towards SLFN11 compared to the tRNA-Met bound structure (Fig. 2D). The interaction with the variable arm could allow SLFN11 to distinguish between type I and type II tRNAs, possibly affecting the affinity for different tRNAs.

Line 269: The authors report that SLFN11 mutants bind ssDNA but not dsDNA. Please elaborate what the consequence of this observation is.

We added the following sentence to the text:

Thus, SLFN11^{S219D} and SLFN11^{T230D} exhibit the same DNA binding properties as SLFN11^{wt}, indicating that phosphorylation at these two residues does not affect SLFN11's ability to bind ssDNA.

Line 281 and 288: The authors report a “significant reduction”, however, do not provide numbers or statistical tests to support this. Please also see major point #2

We rephrased this part, as we did not perform a statistical test to measure significance.

Line 389: As T230 is specific for SLFN11, the site is indeed unique (not just might be unique).

We corrected the sentence to “Finally, T230 is unique to SLFN11, as it is not conserved among other subgroup III Slfn proteins.”

Line 463 and throughout: Please indicate which mask was used to calculate the FSC (presumably cryoSPARC automasking?).

The masks for the calculation of FSC were created by cryoSPARC automasking. The resolution were determined at 0.143 FSC with 3D FSC plots using Remote 3DFSC Processing Server (Tan et al., 2017). This information was added to the method section.

Line 481: “subject” should be “subjected” (also other occurrences thereafter).

We corrected all occurrences of this error in the text.

Figure 1a+b: Please also highlight the location of S219 and T230 in the structure.

We highlighted the location of S219 and T230 in the structure.

Figure 3: Consider to show structures instead of the reconstructions.

We change the displayed panels in the figures and now we show both the reconstruction and the structure of SLFN11 bound to tRNA-Leu in Figure 1B. In Figure 2A, we show the reconstruction of SLFN11 bound to tRNA-Met. Additionally, we show the cryo-EM maps that were used for model building together with the respective structures in Supplementary Fig. 3.

Figure 3B: label where the active sites are (Nuclease active site I and II)

We labelled corresponding nuclease active sites in now Figure 1B.

Figure 3B/C: highlight where C is in B.

We highlighted where the panel C (now F) in the panel B is now in Figure 1B.

SFigure 5+6: This figure could benefit from domain labels (in addition tot he color code).

We added domain labels for both Supplementary Fig. 5 and 6 (now Supplementary Fig. 16 and 18).

SFigure 7C: The legend reports that differences are indicated. However, it does not become clear from panel C what are the differences. Maybe use the same color in different shades for the same structure region to make the comparison more intuitive.

We updated Supplementary Fig. 7 (now Supplementary Fig. 17) to be clearer. Now each structural motif is unified by the colour in all four panels A to D. We also modified the text to properly describe the observed structural changes in more detail.

SFigure 8: The figure contains data that was not mentioned in the text (many different ATPase assays). If the data is relevant, include and describe it in the text. Remove it from the figures otherwise.

The reference and the description of the data is mentioned in lines 295-299.

SF Figure 9: It is not quite clear how processing was done here. Also the method text does not help to understand it. Please clarify this.

We updated the processing scheme in Supplementary Fig. 9 (now Supplementary Fig. 2) together with the corresponding method section to clarify how the processing was done.

- Christian Dienemann (signed review)

Reviewer #3 (Remarks to the Author):

In this work, the authors reveal novel insight into the regulation of the SLFN11 functions and enzymatic activities. They provide cryoEM structures of the SLFN11 phosphomimetic mutant S753D bound to ATP, and of the SLFN11 WT protein bound either to tRNAMet or tRNALeu.

Globally, the study shows that the phosphorylation of S753 prevents the dimerization of SFLN11 and allows the ATP binding. The cryoEM structures with two tRNAs, one with a long variable loop (tRNALeu) and one with a short-one (tRNAMet) give insights into tRNA recognition and cleavage, as well as its regulation by phosphorylation at S219 and T230. Although previous studies have already provided information about the structure of SLFN11 and its interaction with tRNAs, the impact of phosphorylation remains elusive. The manuscript is written in a very clear way. I recommend publication of this article after correction, which would be of high interest for the broad readership of this journal.

We thank the reviewer for the kind appreciation of our work and we hope that we address all issues in the revised version of the manuscript.

I have several suggestions/comments:

- A general remark : it is difficult to judge of the quality of the map because we have no figure with the model fitted in the map. For instance, it could be informative to see the density for the ions, for the active sites... Figures with density should be added either in the main text or in supplementary information.

We now show the cryo-EM densities around the active sites for all four structures we obtained: tRNA-Met is shown in Fig. 2B, tRNA-Leu in Supplementary Fig. 4B, and tRNA-Leu in pre- and post-cleavage states in Supplementary Fig. 6A.

- For structures with tRNA, it is difficult to appreciate in the figures what part of the tRNA is directly bound to the protein. It could be informative to have a figure where we can see the L-shaped tRNA and which arms make contacts with SLFN11. Add the name of the arms on the figures and precise which part of the variable loop is visible. I guess not all the variable loop is observed for tRNALeu.

We now present the side view (B) and bottom view (D) of tRNA-Leu bound to the nuclease domains of the SLFN11 dimer in Figure 1. There we show that the SLFN11 nuclease domains interact with the acceptor stem and the T-arm of the tRNA. The structural features of the tRNA are coloured as in Figure 1C. Also we show the detailed view on the interactions of variable arm of tRNA-Leu with the SLFN11 nuclease domain in Figure 1F. We also visualize which part of tRNA-Leu and tRNA-Met is modelled in Supplementary Fig. 3.

- It is not clear for me why tRNALeu which is a good substrate for SLFN11 is not released after cleavage, so why it is still bound to SLFN11. Could the author provide a gel made in the same condition than the cryoEM sample to be sure that the structure SLFN11/tRNALeu is a post-cleavage structure.

We provide a nuclease gel that represents the conditions that were used for the preparation of the cryo-EM samples (Figure 2C). We used FAM-labelled tRNA-Leu-TAA and tRNA-Met-CAT, instead of the unlabelled tRNA variants that were used for cryo-EM, to visualise tRNA cleavage on a gel. For gel

loading and optimal detection of a fluorescent signal in a gel, we diluted the reaction to reach a final concentration 50 nM of FAM-labelled tRNA. We observed 66 % and 56 % cleavage of tRNA-Leu and tRNA-Met, respectively. Thus, both cleaved and uncleaved tRNA-Leu and tRNA-Met are present in the sample with a majority being cleaved. As we predominantly see cleaved tRNA in our reconstructions, we speculate that SLFN11 is having a higher affinity towards cleaved tRNA than towards uncleaved tRNA. Yet, the *in-vitro* conditions are different from the conditions in living cells, where tRNAs are modified. Furthermore, an unknown interaction partner, which might be missing in the *in vitro* reaction, could help SLFN11 to release the cleaved tRNA.

-Why one site of SLFN11 is cleavage-deficient?

In the cleavage proficient nuclease active site I, additional density indicates the presence of a second metal ion, whereas nuclease active site II shows density for only one metal ion (Fig. 1E and Supplementary Fig. 4B). Further, an examination of nuclease active site I in the pre- and post-cleavage states of the tRNA revealed that, in the pre-cleavage state, there was clear evidence for the second Mn^{2+} ion, whereas in the reconstruction with cleaved tRNA, the density for the second ion vanished (Supplementary Fig. 6B). The vanishing of the second ion after cleavage suggests that the nuclease uses a two-metal-ion mechanism for cleavage, where both Mn^{2+} ions are necessary for catalysis. Active site I, which contains both metal ions, is therefore cleavage-proficient, while active site II, with only one metal ion, is not.

- Were the NanoDSF experiments repeated three times? It would be interesting to have the T_m values with SD.

We performed all NanoDSF experiments in triplicates and added the mean T_i values with SD.

- There is no information about the sequences and production/purification of tRNA samples. This information should be added.

We added a section regarding oligonucleotides in the methods section. As we ordered synthetic oligonucleotides, we did not produce or purify them on our own. The sequences of all oligonucleotides used are provided in Supplementary Table 2.

- The SLFN11 protein is produced in insect cells. Have the authors checked the phosphorylation status of the protein after purification?

In our previous study (Metzner et al., 2022), we investigated the phosphorylation status of SLFN11, which was expressed in insect cells. Mass spectrometry analysis clearly identified tryptic peptides with unphosphorylated S219, T230, and S753. The corresponding phosphorylated peptides were not detected. We also did not observe any other phosphorylated peptides that could hint at the presence of another phosphorylation site. These findings confirm that the unphosphorylated protein is enzymatically active.

- Have you tried to model the structure of SLFN11^{S219D} and T230D using AlphaFold3? Would this help your discussion?

AlphaFold2 and AlphaFold3 were valuable tools for both this work and our previous manuscript (Metzner et al., 2022). Surprisingly, AlphaFold predicts SLFN11 in the 'SLFN11^{S753D} conformation,' which also resembles the conformation of SLFN5. However, the experimentally observed dimeric state of SLFN11^{wt}, as determined by cryo-EM, highlights the necessity of structural biology experiments. Hence, modeling the structure of SLFN11 with single-point mutations (S219D or T230D) did not provide further insights into the roles of these mutations, as AlphaFold3 predicted a SLFN11^{S753D}-like conformation for these mutations as well. Based on our biochemical data, these mutations exhibit the same properties as dimeric SLFN11^{wt}, leading us to conclude that they likely do not induce a conformational change toward an SLFN11^{S753D}-like structure. Therefore, the structures of SLFN11^{S219D} or SLFN11^{T230D} should remain in a state, similar to SLFN11^{wt}.

We thank the editor and all reviewers for their positive feedback on the revised version of the manuscript. We hope that we address all remaining issues in the final version of the manuscript.

Point-by-point response to referees:

Reviewer #1 (Remarks to the Author):

The authors have successfully addressed my concerns. This manuscript is suitable for publication in Nature Communications.

We thank the reviewer for the recommendation of our manuscript for publication.

Reviewer #2 (Remarks to the Author):

The authors have addressed all points raised.

-- Christian Dienemann (signed review)

We thank the reviewer for the kind appreciation of our revised manuscript.

Reviewer #3 (Remarks to the Author):

The authors have responded to all my points and corrected their paper as requested. I recommend publication of this paper in its revised version.

We thank the reviewer for the recommendation of our revised manuscript for publication.